# Hidden chaos factors inducing random walks which reduce hospital operative efficiency

**Antonio Javier Rodríguez-Hernández[1], Carlos Sevcik[2¤]***

**1** Hospital La Fuenfría, Carretera de Las Dehesas, Madrid, Spain, **2** Centro de Biofísica y Bioquímica, Instituto Venezolano de Investigaciones Científicas, Caracas, Venezuela

¤ Current address: SciMeDAn, Barcelona, Spain
* carlos.sevcik.s@gmail.com

**Data Availability Statement:** All relevant data are within the manuscript and its Supporting information files. The Command panels data used in this study are provided as Supporting information

## Abstract

Operative parameters of La Fuenfría Hospital such as: hospitalized patients; daily admissions and discharges were studies for the hospital as a whole, and for each hospital's service unit (henceforth called '*services*'). Conventional statistical analyzes and fractal dimension analyzes were performed on daily In-Patient series. The sequence of daily admissions and patients staying on each service were found to be a kind of random series known as *random walks* (*Rw*), sequences where what happens next, depends on what happens now plus a random variable. *Rw* analyzed with parametric or nonparametric statistics may simulate cycles and drifts which resemble seasonal variations or fake trends which reduce the Hospital's efficiency. Globally, inpatients *Rw* s in LFH, were found to be determined by the time elapsed between daily discharges and admissions. The factors determining LFH R were found to be the difference between daily admissions and discharges. The discharges are replaced by admissions with some random delay and the random difference determines LFH *Rw* s. These findings show that if the daily difference between admissions and discharges is minimized, the number of inpatients would fluctuate less and the number of unoccupied beds would be reduced, thus optimizing the Hospital service.

## Introduction

The efficiency of a service provider institution depends largely on avoiding fluctuations between active and inactive periods. This is true for hotels, airplanes companies, hospitals or any manufacturing production lines. Control and predictability rationalize costs and inventory expenses, this has long been known to proponents of total quality control management [1].

In general, hospitals are service institutions which receive a random queue of service demanders (patients) whose admissions depends both on patient availability and on the hospital's capacity to attend or admit a patient. In a hospital operating at full capacity a patient must be discharged prior to the admission of a new patient. If a hospital is part of a network of hospitals, the situation becomes even more complex, since it depends on the hospital'scommunication between the discharge and admission and discharge sections/systems, and that of hospitals in the network which transfer patients to another hospital. In the latter situation

and are freely available. A short Readme text file is also included, which explains the LFH data file.

**Funding:** All funds were provided by Hospital La Fuenfría, Servicio Madrileño de Salud (SERMAS), Comunidad Autónoma de Madrid, Madrid, Spain.

**Competing interests:** No conflict of inter for any author.

transportation between hospitals may be another factor dependent on availability of means of transport, or if this depends on the patient's themselves, on whether those resources are available to the patient, immediately or with some sort of delay. All these factors conform a complex set of circumstances that determine hospital efficacy, and introduce complex non linearities in the system, and yet, to our knowledge, this non-linearities are never considered when hospital efficiency is evaluated.

Non linear systems such as those described in the preceding paragraph, are poor candidates for analysis with standard statistical means. This has been shown for may events such as earthquakes, weather, adult and fetal hearth beat, epidemics, highway traffic, stock market fluctuations, machine failures and so on [2–10]. Still, the common practice to evaluate an detect institutional efficiency (or lack of it) is generally limited to actuarial studies and certain descriptive statistics loaded into administrative data bases. Thus, modifications are made to b budgets, personnel hiring, firing, promotions, demotions, and even hospital closures, may be based on these questionable analyzes. This paper describes the analysis of a real hospital, and benefits from the development of a simple algorithm [11] to estimate the Hausdorff-Besicovitch [12, 13] dimension using a kind of box-counting algorithm, to estimated fractal dimension, called with increasing frequency as "Sevcik's dimension" [14–19]. Thus, from this paper on, we will start calling "Sevcik's fractal dimension": $D_s$ since this avoids some formal mathematical confusions with the Hausdorff-Besicovitch dimension. [ ©2018 Google LLC All rights reserved. **GogScholar** and the **Google** logo are registered trademarks of **Google LLC**.] [20] lists 170 cases of fractal analysis, in many fields, up to November 2, 2021 [21] with this [11] algorithm.

La Fuenfría Hospital (LFH, Servicio Madrileño de Salud, SE*R*MAS) is a public mid and long term stay hospital [in the Madrid's (Spain) Autonomous Region] (Comunidad Autónoma de Madrd, in original Spanish) for chronic patients. Spain has a life expectancy of 83.61 years (the 6[th] longest in the world), increasig at a rate of 0.210% per year) [22]. This fast growth makes Spain a candidate to become the country with world's longest life expectancy in the near future.

In 2019, Spain had a population of 46,738,578 inhabitants, and 26.5% ($\approx$ 12.2 million)) were above 60 years of age (calculated based on [22–25]). One would hence expect that hospitals for chronic diseases would have a significant and constant demand. But, the curve of inpatients when the curve of patients staying al the Hospital (called inpatients, abbreviated **InP**) during the period under study (Fig 1A) fluctuated between being completely full followed by periods with only $\approx$ 52% occupancy (the hospital has 192 beds). Analyzed with classical administrative and statistical tools, this behavior could be attributed to some type of service demand, seasonal periodicity or, even worse, attributed to administrative deficiencies. Regardless, of the explanation is given, this behavior is undesirable, leading to reduced hospital efficiency, increased costs, and thus, the the underlying cause must be identified and eliminated as thoroughly as possible.

Here we use fractal analysis, Fourier transforms, spectral analysis, and conventional statistics, to show that small hidden factors may indeed have a dramatic impact in reducing the hospital's operative efficiency. The purpose of this paper is to show that factors that are not usually considered in hospital performance analysis may have a huge impact on hospital efficacy.

## Methods

### Statistical methods and spectral density (SD) calculation

**Statistical methods.** Medians and their 95% confidence interval (CI) were calculated using the nonparametric Hodges and Lehman method [26]. Random distribution about the

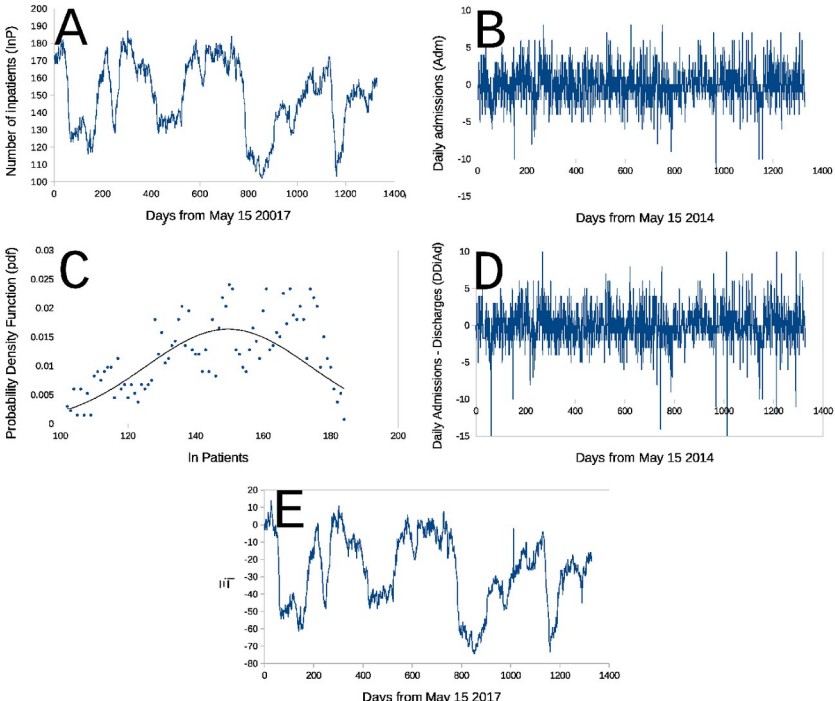

**Fig 1. Daily in-patients (InP) at La Fuenfría Hospital from May 1 2014 and December 19 2017.** Panels are: **Panel A**- Number of inpatients (**InP**) at La Fuenfría Hospital per day during the study duration; **Panel B**- Daily variations of inpatients (**DInP**) at the Hospital's [$\Delta y$ in Eq (2)]; **Panel C**- Hospital's daily stays (Panel A) **p**robability **d**ensity **f**unction (*pdf*, the probability of observing a given mensurable or enumerable random event) for any number of inpatients (**InP**) staying at the Hospital observed during the study period (•) [28], a Gauss line ($N[\overline{y}, s[y]]$) with the same mean and variance ($s[y]^2$) as the set of 1329 data points making the series in Panel A ($\overline{y} = 149.2$, $s[y] = 24.4$); **Panel D**- Daily difference between admissions (Fig 2A) and discharges (Fig 2C) in the Hospital (**DDiAd**). **Panel E**- Random walk $\Xi_i$ built for **DDiAd** (Panel D) using Eq (5), please note the extreme resemblance to the sequence of **InP** in Panel A. In all cases abscissa is the number of days after May 1 2014. The largest negative or positive peaks are off scale outliers.

median ($\hat{y}$) was analyzed with the above and below *runs* (one or more data appearing as a sequence above or below the median), following the methods of Mood [27] and Wilks [28, 29]. Gaussianity (also known as normality) was verified with the Jarque-Bera [30], the robustified Jarque-Bera-Gel [31] and the Shapiro Wilk [32] tests. Linear regressions were carried out using the Theil nonparametric method [26] and correlations were estimated using the Spearman rank correlation coefficient [26]. Sequences were compared with the nonparametric Smirnov [33] test based on Kolmogorov statistics [34]. The procedures to generate random sequences to verify the differences between their fractal dimension which were described elsewhere [11, 35–37]

**Fast Fourier Transforms (FFT) and calculating spectral density.** FFT were calculated with a C++ (using g++ version 5.4.0 20160609 with C++14 standards, www.gnu.org) implementing the Cooley and Tukey algorithm [38], with Hamm windowed data [39]. Spectral densities were then obtained from the real and imaginary components of the FFT [39].

## Building the time series

Series were built for La Fuenfría Hospital's admissions, discharges (**Dis**) and hospitalized patients, called "inpatients", and shortened to **InP** in this paper. The Hospital depends on the Public Health System Network [Servicio Madrileño de Salud (SERMAS)] of the Autonomous

Madrid Region, which, like most in Spanish autonomous regions, requires that all hospital information in the area is stored into a central data base. In the mid-term stay hospital this information is kept using a program called SELENE©. SELENE© is jointly developed by UTE Siemens– INDRA [40]. SELENE© keeps data in a central database for SERMAS. Data were retrieved from SELENE© as monthly spreadsheets for each kind of information, and were integrated in a single file covering our entire period of study for each type of information.

To build the sequences for this study, we used data collected monthly as SELENE© Hospital's spreadsheet files called "*Cuadros de Mando*" which we translate *Command panels*. In them, information appears as daily data for La Fuenfría Hospital from 5/1/2014 up to 12/31/2014, then they run from January 1 to December 31 for years 2015 and 2016, and from 1/1/2017 up to 12/19/2017.

Data was routinely hospital's statistics, was not recorded purportedly for this study, and we found transcription errors in the SELENE©-exported data base, which were dealt with as follows:

- January 1$^{st}$ data corrections. An obvious error detected in the time series was that more than 900 admissions or discharges appeared for 1/1/2015 and 1/1/2016, which is impossible for a hospital with 192 beds. The source of such errors appears to be a bug in the software used to gather data which did not evaluate the first datum of the year correctly. Those points were replaced by averaging December 31 of the previous year with January 2 of the current year, respectively. This error affected 2 out of 1329 days, or $\approx 0.16\%$ of data.

- **Missing data corrections**. In Admissions (**Adm**) and Discharges (**Dis**) series we found a gap without information from 5/29/2014 to 6/6/2014 (both inclusive), in daily in–patients. This gap (9 out of 1329 days, or $\approx 0.67\%$ of data) was closed by filling the *data on daily changes in the sequences for the gap days* with a same value calculated as

$$\{y_{(6/6/2014)} \leftrightarrow y_{(5/29/2014)}\} = \frac{y_{(6/7/2014)} - y_{(5/28/2014)}}{9}, \tag{1}$$

where dates follow the "mm/dd/yyyy" style, and the double headed arrow at the left of Eq (1) points at the limits of the gap to be filled. In Eq (1), as well as elsewhere in this paper, brackets are used to indicate sets.

- Daily differences between **Adm**, **Dis** and **InP**. Once daily data admissions (**DAdm**), daily discharges (**DDis**) and daily inpatients (**DInP**) sequences were built, new series were developed by subtracting from each daily value, the immediately preceding day's similar datum. A set of sequences of one-day differences

$$\Delta y_t = \begin{cases} 0 & \Rightarrow t = 0 \\ y_t - y_{t-\Delta t} & \Rightarrow t > 0 \end{cases} \tag{2}$$

with $t = 0, 1, \ldots, 1328$ days, was built. In Eq (2) each $t$ is a given day, and $\Delta t = 1$ is a one-day time difference, in which the first day was arbitrarily taken as 0. The procedure was repeated for admissions, discharges and stays, to obtain the corresponding daily differences sequences for the three kinds of parameters shown in Fig 2A and 2B.

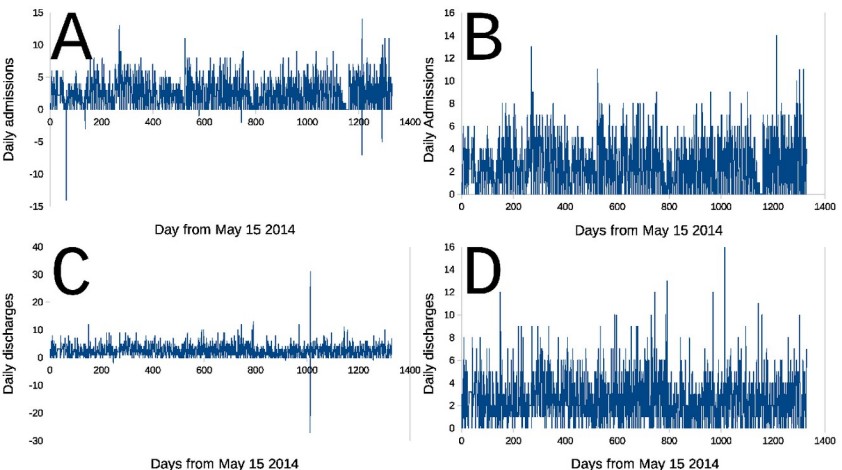

**Fig 2. Sequences of daily admissions and daily discharges from La Fuenfría Hospital.** Panels are: **A**- Daily admissions (**Adm**) sequence; **B**- Same as panel A, but scale modified to only to show data ⩾ 0; **C**- Daily discharge (**Dis**) sequence; **D**- Same as panel C, but the scale is modified to only show data ⩾ 0. In all cases abscissa is days elapsed since May 1 2014, Ordinate is number of daily admissions or discharges determined as indicated in **Building the time series** in **Methods**. Negative peak represent transcription errors of the data made available for this study, perhaps positive outliers could have the same origin, however this is harder to demonstrate.

## Ethics statement, author's conflicts of interests and privacy of the information used in the study

No human or animal subjects were used in this study. Thus no animals were sacrificed. Neither personal nor animal pain or discomfort were induced at any point of the study.

In compliance with the European Data Protection Law [41] no patient's personal or clinical information is revealed in this paper. La Fuenfría Hospital Ethics Committee approved this study as indicated in a letter submitted to PLoS One.

Though it is the author's interest always to improve the efficiency of LFH. The authors state that they have no conflicts of interest or competing interests in the subject of this study.

Excluding personal or clinical patient information, everything stored in the SELENE system, including the La Fuenfría Hospital "Cuadros de Mando" which we translated here as "Command panels", are public information according to Spanish Law. All the original La Fuenfría Hospital "Cuadros de Mando" which we translate "Command panels" spreadsheets cis be provided with this revised manuscript. There is also a short Readme text file explaining some issues of the LFH data file. The Command panels, and a short Readme text file, used here are provided to PLoS One, and are freely available to anyone interested in them.

## Data availability and privacy of the information used in the study

All relevant data are within the manuscript and its S1 Data files. The Command panels data used in this study are provided as S1 Data and are freely available. A short Readme text file is also included, which explains the LFH data file.

Excluding personal or clinical patient information, everything stored in the SELENE system, including the La Fuenfría Hospital "Cuadros de Mando" which we translated here as "Command panels", are public information according to Spanish Law. All the original La Fuenfría Hospital "Cuadros de Mando" which we translate "Command panels" spreadsheets cis be provided with this revised manuscript. The translated Command panels are provided to PLoS One, and are freely available to anyone interested in them.

LFH Command Panels also include names and ID codes of Hospital personnel for administrative purposes. Since these are also protected by Spanish and EU laws on personal information privacy they were remove prom data posted in PLoS ONE.

### Ethics statement and author's conflicts of interests

No human or animal subjects were used in this study. Thus no animals were sacrificed. Neither personal nor animal pain or discomfort were induced at any point of the study.

In compliance with the European Data Protection Law [41] no patient's personal or clinical information is revealed in this paper. La Fuenfría Hospital Ethics Committee approved this study as indicated in a letter submitted to PLoS ONE.

Though it is the author's interest always to improve the efficiency of LFH. The authors state that they have no conflicts of interest or competing interests in the subject of this study.

## Results

### Analysis of runs above and below the median

During the period studied, the number of hospital in–patients were 150 (149–151) (median, $\hat{y}$, and 95% CI, n = 1329 days) ranging (102 –- 187) patients, meaning that Hospital's is 0.77 (0.53–0.77) (median, $\hat{y}$, and 95% CI). Let us consider a random variable $y = f(t)$ such that $f(t)$ is independent from $f(t − \Delta t)$ or from $f(t + \Delta t)$ (where $\Delta t$ is a short time lapse, ideally $\Delta t \rightarrow 0$). This is, a variable for which its present is independent from its past, and whose present does not determine its future. Such variable produces a sequence of data which are uniformly distributed around their median $\hat{y}$. We say that a run occurs when one or more sequence points fall above or below the median, this is easily grasped by looking at Fig 3. In Fig 3C there are 5 runs, and in Fig 3D there are 9 runs. Points that are equal to $\hat{y}$ are not considered in the analysis. Fig 3 presents four sets of monthly averages from the **F**unction **R**ecovery **U**unit (**FRU**). Fig 3A, results from the total of inpatients (**Adm**) sequence ($n = 22$, $n_{runs} = 11$, ***probability of random run distribution about the median*** $P_{rnd}$ is $\approx 0.15$); Fig 3B, Total discharged patients' (**Dis**) sequence ($n = 22$, $n_{runs} = 8$, $P_{rnd} \approx 0.04$); Fig 3C, is the sum of lengths of stay (**LoS**) for all in–patients (**InP**) in the Unit during each month ($n = 23$, $n_{runs} = 5$, $P_{rnd} \approx 4.5 \cdot 10^{-4}$); Fig 3D, is a sequence of the average stay duration (Mean **LoS**) of each in–patient at the Unit ($n = 23$, $n_{runs} = 10$, $P_{rnd} \approx 0.13$); Fig 3E, during de observation period the Function Recovery Unit gained relevance and the number of beds were increased, panel shows the changes in beds at the beginning in several months, as expected, the analysis of this sequence was found to be non random ($n = 23$, $n_{runs} = 5$, $P_{rnd} \approx 4.5 \cdot 10^{-4}$), actually $n_{runs}$ may be just 3 in this case, but the algorithm probably found tiny decimal differences between data Fig 3 an $\hat{y}$ in one segment; if $n_{runs} = 3$, should n runs be 3 then $P_{rnd} \ll 4.5 \cdot 10^{-4}$. $P_{rnd}$s are probabilities (obtained with the Wald–Wolfovitz test) that the sequences are not randomly distributed about their medians. In all cases abscissa is the month number during the study period (January 2016 = 1, November 2017 = 23). Out of these $P_{rnd}$ values the only one indicating a statistically significant non zero number of runs, occurs for **InP** sequence in Fig 3C [42, 43]. The sequence non-randomness the Fig 3C could have several reasons, one of them could be some kind of 'periodicity' (which would be odd, since its period would be longer that 2 years), but it could also stem from some kind of *random walk*. During the period studied, the Hospital's in–patients were 150 (149–151) (median, $\hat{y}$, and 95% CI) ranging (102–187) patients. If we consider a random variable $y = f(t)$ such that $f(t)$ is independent from $f(t − \Delta t)$ or from $f(t + \Delta t)$ (where $\Delta t$ is a short time lapse, ideally $\Delta t \rightarrow 0$). In other words, a variable for which its present is independent from its past, and whose present does not determines its future. Such variable produces a sequence of

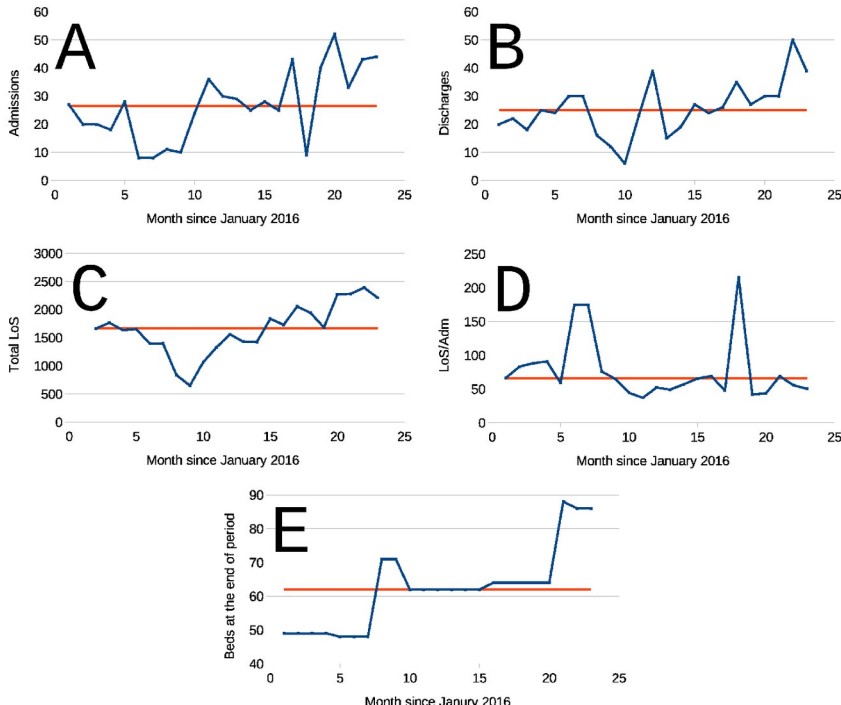

**Fig 3. Four examples of runs around the median ($\hat{y}$) from La Fuenfría's Function Recovery Uunit.** (**FRU**). **A**: Total admissions (**Adm**) sequence ($n = 22$, $n_{runs} = 11$, $P_{rnd} \approx 0.15$); **B**, Total discharges (**Dis**) sequence ($n = 22$, $n_{runs} = 8$, $P_{rnd} \approx 0.04$); **C**, is the sum of lengths of stay (**LoS**) for all in–patients (**InP**) in the Unit during each month ($n = 23$, $n_{runs} = 5$, $P_{rnd} \approx 4.5 \cdot 10^{-4}$); **D**, is a sequence of the average stay duration (Mean **LoS**) of each in–patient at the Unit ($n = 23$, $n_{runs} = 10$, $P_{rnd} \approx 0.13$); **E**, throughout the observation period the Function Recovery Unit gained relevance and the number of beds were increased, the panel shows the changes in beds at the behind ib several months, as expenses, the analysis of this senescence was found to be non random ($n = 23$, $n_{run} = 5$, $P_{rnd} \approx 4.5 \cdot 10^{-4}$, actually $n$ runs may be just in this case, the algorithm probably found very minor decimal differences between data an $\hat{x}$ in one segment; if n $runs = 3$, should n runs be 3 than $P_{rnd}$ $4.5 \cdot 10^{-4}$. $P_{rnd}$ s are probabilities (obtained with the Wald–Wolfovitz test) that the sequences are randomly distributed about their medians. As indicated in the text of this communication $P_{rnd}$ stands for probability of random distribution about the median. Fluctuations in **E** resulted from administrative decisions, thus a low $P_{rnd}$ was to be expected. In all cases abscissa is time expressed as month number in the sequence, (January 2016 = 1, November 2017 = 23).

data which are uniformly distributed about their median $\hat{y}$. We say that run occurs when one or more sequence points fall above or below the median, this is easily grasped by looking at Fig 3. In Fig 3C there are 5 runs, and in Fig 3D are 9 runs. Points that are equal to $\hat{y}$ are not taken into account in the analysis. Fig 3 presents four sets of monthly averages from the **Function Recovery Uunit** (**FRU**). Fig 3A, results from the total of inpatients (**Adm**) sequence ($n = 22$, $n_{runs} = 11$, **probability of random run distribution about the median** $P_{rnd}$ is $\approx 0.15$); Fig 3B, Total discharged patients' (**Dis**) sequence ($n = 22$, $n_{runs} = 8$, $P_{rnd} \approx 0.04$); Fig 3C, is the sum of lengths of stay (**LoS**) for all in–patients (**InP**) in the Unit during each month ($n = 23$, $n_{runs} = 5$, $P_{rnd} \approx 4.5 \cdot 10^{-4}$); Fig 3D, is a sequence of the average stay duration (Mean **LoS**) of each in–patient at the Unit ($n = 23$, $n_{runs} = 10$, $P_{rnd} \approx 0.13$); Fig 3E, along de observation period the Function Recovery Unit gained relevance and the number of beds were increased, panel shows the changes in beds at the benign in several months, as expected, the analysis of this sequence was found to be non random ($n = 23$, $n_{runs} = 5$, $P_{rnd} \approx 4.5 \cdot 10^{-4}$), actually $n_{runs}$ may be just 3 in this case, the algorithm probably found very minor decimal differences between data Fig 3 an $\hat{y}$ in one segment; if $n_{runs} = 3$, should n runs be 3 then $P_{rnd} \ll 4.5 \cdot 10^{-4}$. $Ps$ are probabilities (obtained with the Wald–Wolfovitz test) that the sequences are not randomly distributed

about their medians. Fluctuations in Fig 3E, resulted from administrative decisions, thus a low $P_{rnd}$, indicative of non randomness, was to be expected. In all cases abscissa is month number in the study period (January 2016 = 1, November 2017 = 23). Out of these $P_{rnd}$ values the only one indicating a statistically significant non zero number of runs, occurs for **InP** sequence in Fig 3C [42, 43]. The non-randomness of sequence Fig 3C could have several reasons, one of them is some kind of 'periodicity' (a strange one, since its period would be longer that 2 years), but it could also stem from some kind of *random walk* determined by the uncertainties in $n_j$ and **$LoS_i$, $j$** [11, 37, 44]. It is difficult to prove, but the apparent 'seasonality' was probably determined by the increase in Unit's bed capacity depicted in panel Fig 3E. The figure illustrates that relatively simple sequences of random variables having unknown pdfs such as $n_j$ and **$LoS_{i,j}$**, may interact in strange manners to produce apparently less random "simi-periodic" sequences such as Fig 3C. determined by the uncertainties in $n_j$ and **$LoS_i$, $j$** [11, 37, 44]. It is difficult to prove, but the apparent 'seasonality' was probably determined by the increase in Unit's bed capacity depicted in panel Fig 3E. The figure illustrates that relatively simple sequences of random variables having unknown pdfs such as $n_j$ and **$LoS_{i,j}$**, may interact in strange manners to produce apparently less random "semi periodic" sequences such as Fig 3C.

**LaN Fuenfría Hospital data statistical characteristics.** Fig 2 presents sequences built in the preceding **Building the time series** Subsection. Panels Fig 2A (**Adm**) and Fig 2C (**Dis**) are sequences with a few negative, artefactual (see **Building the time series** Subsection, outliers. Panels Fig 2B and 2D are the same data with the scale magnified to see the positive part and better appreciate data detail when $y_t \geqslant 0$.

Testing with the Jarque-Bera [30], the robustified Jarque-Bera-Gel [31] and the Shapiro-Wilk [32] tests, we found that none of the data sequences collected during the study period from La Fuenfría Hospital, were Gaussian (also called normal) variables, (Probability of Gaussianity, $P \ll 10^{-6}$). **InP** sequences contained a large number of ties (points with same value). In ti **InP** sequence every value is repeated

$$\frac{n_{total}}{InP_{max} - InP_{min}} = \frac{1329}{187 - 102} \approxeq 16 \text{ times.}$$

In the expression: **$InP_{max}$**, is **InP**'s maximum values; **$InP_{min}$**, is **InP**'s minimum values and $n_{total}$, is total number of points in the sequences. Given the narrow range there are only $187 - 102 = 85$ possible different values among 1329 integer data in the series, they are not all equally likely. Please notice that the number of ties increases as **$InP_{range}$** $\to 0$, that is the more uniform bed occupation gets, the larger the number of ties, in the **InP** sequence, will get. This is to be expected when integers are sampled in a narrow range, under these conditions any statistical test, parametric or not, must be taken with caution, since test power decreases wit ties. Tie corrections were used for the nonparametric statistical procedures employed here [26]. Since the Hospital has 192 beds, data indicate that the Hospital operates with some leeway, having an occupancy ranging from 53.1 to 97.3%, with a 78.1% median.

With the data in the sequence of **InP** (shown in Fig 2A) and Eq (2) a curve of **DInP** at the Hospital was built, this is shown in Fig 1B. The sequence in Fig 1D was obtained by subtracting to each **Adm** the same day **Dis**. Fig 1D presents a sequence obtained by subtracting **Dis**—**Adm** (**DDiAd**) to the Hospital.

A direct visual comparison between sequences in Fig 1B (**DInP**) and Fig 1D (**DDiAd**) shows, that in spite of some differences, they are similar, and very different from the series in Fig 1A (**InP**). Sequences in Fig 1B (**DInP**) and Fig 1D (**DDiAd**) compared with the Smirnov test had the same distribution, that is, their distributions are statistically

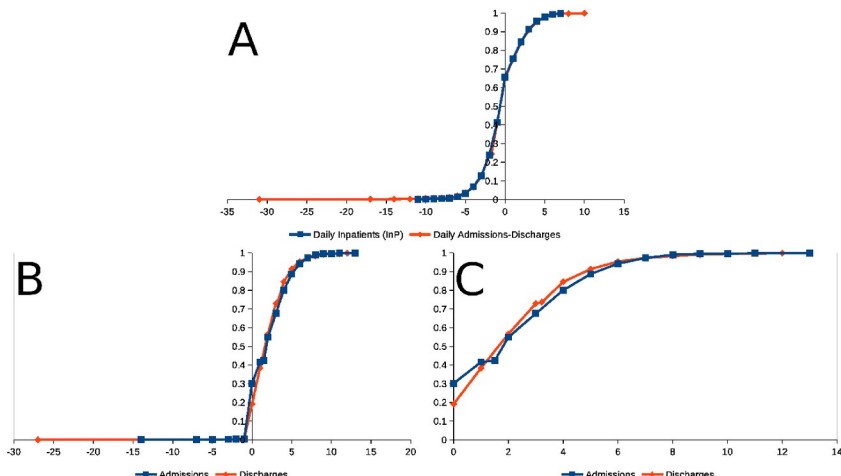

**Fig 4. Empirical Probability Distribution Functions (PDF)** [28] **of daily admissions and stays at La Fuenfría Hospital from May 12014 to December 19 017. Panel A**: Blue ine with ■ PDF of LFH in-patients (**InP** in Fig 1A); Red line with ♦, PDFs for daily differences in admissions (**Adm**) and discharges (**Dis**), called **DDiAd** in Fig 1D. **Panel B**: Red line with ♦ PDF of LFH daily Admissions in LFH (called **Adm** in Fig 2A); Red line with ■ PDF for LFH daily Discharged **Dis** in Fig 2C). **Panel C**: Same as Paanel B, but with abscissa expanded to exclude outliers. In all cases abscissa is the value ve aare interested in, and ordinate is the estimated accumulated probability from $-\infty$ up to the abscissa value.

indistinguishable (Probability of identity $P \lessgtr 1$, $n = 1329$, two tails). Yet, when Smirnov test was used to compare sequences Fig 2A and 2C, admission (**Adm**) and discharges (**Dis**) sequences were found to be highly statistically significantly different (Probability of identity $P \ll 10^{-6}$, $n = 1329$, two tails).

Empirical Hospital occupancy Probability Distribution Functions (PDF) [28] calculated from these data is shown in Fig 4. Daily hospital occupancy data set was found to be not Gaussian when tested with the Jarque-Bera [30], and subsequently with the robustified Jarque-Bera-Gel [31] and the Shapiro Wilk [32] tests ($P \ll 10^{-6}$, in the three cases). The perception of data lacking Gaussianity becomes intuitive when very disperse data (dots) are observed around a Gauss pdf line in the same panel. The Gauss pdf line (shown also as pdf, in Fig 1C) with the same mean ($\bar{y}$) and variance [$s^2(y)$] as the set of 1329 data points forming the sequence in Fig 1A ($\bar{y} = 149.2$, $s[y] = 24.4$).

**La Fuenfría Hospital daily admissions and discharge variations sequences.** Fig 4 contains empirical PDF [45] calculated for inpatients (**InP**), admissions (**Adn**), discharges (**Dis**) and difference between **Adm** and **Dis**: (**DAdDis**). It is important to note that **Adm** and **Dis** were *were logged daily as separated entries* in Hospital's Command Panels. Data in Fig 4 (especially the expansion presented in Fig 4C) indicates that daily **DInP** is distributed following the *same PDF as DAdDis*, however **Adm** and **Dis** have *different PDFs*. The *largest difference* is observed for probabilities of *no admissions* (blue line in Fig 4C) and *no discharges* occur (red line in Fig 4C) in a given day (at zero value in the abscissa in Fig 4C), it is a day without admissions 30% likely but a day without discharges is only 20% likely. When the abscissa is $> 0$, the probability of daily admissions is surpassed by daily discharges. The largest difference between the curve predicted determines the significance of the difference between two PDFs by using the Smirnov test [33, 46] (based on Kolmogorov statistics [34]). Yet when analyzing at Fig 4C, it is evident that this is not the only difference between the curves.

## Daily fluctuations in stay duration and average stay duration in operative units at La Fuenfría Hospital

**Random fractals.** Two curves are particularly relevant for the analysis of sequences in Figs 1 and 2, these are the, so called, white and Brownian noises [44]. In white noise every point in the sequence is independent from the past, and does not determine the future of the system. In brown noise, every point is dependent on the past and determines the future, although in in unpredictable manners.

*White noise properties.* White noise is a sequence of independent random variables, no value is predictable nor dependent on on any prior or posterior value in the sequence. White noises have a fractal dimension $D_h$ = 2 [11, 37, 44]. White noise curve looks very similar to sequences in Fig 1B and 1D, the random variable producing each point may have diverse PDFs, white noises are randomly distributed about 0, when $N \rightarrow \infty$ white noise variables in a sequence produce $\frac{N}{2} + 1$ runs [47]. When subject to Fourier transformation [48] white noises have a constant spectrum for all frequencies between 0 and $\infty$, reminding *white light* which contains all visible electromagnetic frequencies, leading to call this noise *white*.

*Brownian noise properties.* Brownian noise is also called *Brown noise* [49, 50], *red noise* or *random walk (Rw)* [44]. For Brownian it noise always holds that

$$f(x) = \begin{cases} 0 \text{ if } x = 0 \text{ (a convention for th observation's origin)} \\ f(x - \Delta x) + \upsilon(\mu = 0, \ \sigma^2), \end{cases} \tag{3}$$

where $\upsilon(\mu = 0, \sigma^2)$ is a random variable with mean $\mu = 0$ and variance $\sigma^2$. Variable $\upsilon$ may be a uniform U[-1, 1] pdf, like

$$U[-1, 1] = \begin{cases} 0 & \text{if } x < -1 \\ 0.5 & \text{if } -1 \leqslant x \leqslant 1 \\ 0 & \text{if } x > 1. \end{cases} \tag{4}$$

Yet, very often the variable is a Gaussian (also called *normal*) of the kind $N(0, \sigma^2)$ having mean $\mu = 0$ and variance $\sigma^2$ [11, 44]. Brown noise is a particular case of Markov chains [51], specially important in the *waiting time theory* witch studies situations with repressed queues waiting to receive some service [52, 53]. In layman's terms, Brownian variables strictly depend on their past, yet they are random, it is impossible to predict their future or estimate their past values. Nonetheless, their current value depends on their past, and their future value depends on the present.

Brown noise in hospital parameters is not unexpected, the admission of a patient depends on bed availability, the number of beds depends on many criteria related to the public health system. Availability also depends on demand, the disease suffered by patient occupying a bed, the efficiency of doctors in the hospital, treatment availability and so on. Brownian noise is characterized by wide and slow meandering, and has fractal dimension $D_h$ = 1.5 [Eq (11)] and so does $D_s$ [Eq (14)] when $N \rightarrow \infty$ [11, 44]. Brownian noise distributes around its median (which is not necessarily 0) producing a variable number of runs $\ll \frac{N}{2} + 1$.

**Fractal analysis theory of sequences studied.** As shown in **Methods** and in **Results**, in contrast with other methods to estimate the fractal dimension $D_h$, Eq (14) is extremely easy to calculate and certainly converges towards $D_h$ as $N \rightarrow \infty$, although nobody knows how far infinity is for certain [54], we all belie it is far away. Fractal series studied here have an extent which is hard to classify. An $N$ = 1329 points may be long or short depending on persons's point of view, it seems large for conventional statistics, yet is short when considered in the

context of fractal analysis [37]. A $D_s$ value deduced from a 1329 points series underestimates sequence $D_h$ significantly. There is four-step solution for this:

1. Generate a significant number of sequences with the characteristics of interest using Monte Carlo [55] simulation.

2. Calculate their $D_s$ and var($D_s$) [Eqs (14) and (15)] [11].

3. Calculate the set of traces' mean $D_s$ and its variance including variance between simulated traces as indicated in [11, 35–37].

4. Compare real and simulated sequences parameters using the Vysochanskij–Petunin inequality (details Theorem 1 and in [35, 37]).

## Fractal analysis of sequences from Hospital

**Ruling out sources of errors in sequences.** To analyze data in Table 2 we must consider the transcription errors in the SELENE® data base and how were they corrected for this study, as indicated in **Methods** and the problem of ties mentioned in the **La Fuenfría Hospital data statistical characteristics** Sub-Subsection, also the narrow data rage determines further underestimations of $D_h$.

A case random noises in seen in [37] where much longer sequences needed for $D_s$ to converge towards $D_h$, these sequences contain only decimal digits $\{0, 1, \ldots, 8, 9\} \in \mathbb{Z}$, not real numbers $\{-\infty < x < +\infty\} \in \mathbb{R}$. Empiric data on Tables 1 and 2, may differ from $D_s$ of a calibration white noise from a longer sequences.

**Estimates of the fractal dimension.** $D_s$ values estimated for Hospital sequences in Figs 1 and 2 are given in Table 1. White noise standard set of 100 series with 1329 point s had an estimated $D_s = 1.5983 \pm 0.0104 \left[\overline{D}_s \pm s(\overline{D}_s)\right]$; whereas the Brown noise standard set of 100 series with 1329 points had $D_s = 1.2885 \pm 0.0384 \left[\overline{D}_s \pm s(\overline{D}_s)\right]$.

**Table 1. Fractal dimension $D_s$ estimated for La Fuenfría Hospital sequences.**

| Sequence | $D_s$ | $\sqrt{\mathbf{var}[D_s]}$ |
|---|---|---|
| Admissions (**Adm** | 1.49725 | 0.00356 |
| Discharges (**Dis** $D_s$) | 1.41904 | 0.00400 |
| In Patients (**InP** | 1.33806 | 0.00320 |
| InP variation (**DInP** | 1.56432 | 0.00302 |
| Adm-Dis (**DDiAd**) | 1.43369 | 0.00401 |
| $\Xi_i$ $Rw$ | 1.33922 | 0.00382 |
| Brownian noise§ | 1.28855 | 0.03837 |
| White noise§ | 1.59825 | 0.01039 |

**Dimension $D_s$** calculated with Eq (14); $s(d) = \sqrt{\mathbf{var}(\mathbf{D}_s)}$ calculated with Eq, (15), for all sequences $N = 1329$. **Labels mean: InP**, in patients at La Fuenfría Hospital (Fig 1A); **Adm**, admissions to the Hospital (Fig 2A); **Dis**, discharges from La Fuenfría Hospital (Fig 2D); **InP**, daily sequence of inpatients (Fig 1A); **DInP**, InP daily variation (Fig 1B); **DDiAd**, Adm-Dis (Fig 1D); $\Xi_i$, $Rw$ built for **DDiAd** curve in Fig 1E using Eq (5); **Brownian noise**, Brownian noise calculated as indicated in the **Results' Random fractals** Sub-Section and Eq (7). **White noise**, White noise calculated as indicated under **Random Fractals** Subsubsection.

§ superscript, indicates that $D_s$ is the average of $M = 100$ sequences with $N = 1329$ evaluated with Eq (9), total variance was calculated with Eq (18).

**Table 2. Statistical comparison between fractal dimensions ($D_s$) estimated for the sequences appearing in Table 1.**

|  | Dis | Adm | DInP | $\Xi_i$ | Bro$^\S$ | Whi$^\S$ |
|---|---|---|---|---|---|---|
| InP | $2 \cdot 10^{-3}$ | $4 \cdot 10^{-4}$ | $2 \cdot 10^{-4}$ | $\frac{1}{6} \leqslant P \leqslant 1^\dagger$ | **0.269** | $7 \cdot 10^{-4}$ |
| Dis | ... | $2 \cdot 10^{-3}$ | $5 \cdot 10^{-4}$ | $2 \cdot 10^{-4}$ | $0.039^\ddagger$ | $2 \cdot 10^{-3}$ |
| Adm | ... | ... | $2 \cdot 10^{-3}$ | $5 \cdot 10^{-4}$ | $0.015^\ddagger$ | $0.0053^\ddagger$ |
| DInP | ... | ... | ... | $2 \cdot 10^{-4}$ | $9 \cdot 10^{-3\ \ddagger}$ | $0.045^\ddagger$ |
| $\Xi_i$ | ... | ... | ... | ... | **0.257** | $8 \cdot 10^{-4}$ |
| Bro$^\S$ | ... | ... | ... | ... | ... | $7 \cdot 10^{-4}$ |

**Statistic significances**: calculated using Eq (29). **Labels mean: InP**, in patients at La Fuenfría Hospital (Fig 1A); **Adm**, admissions to La Fuenfría Hospital (Fig 2A); **Dis**, discharges from La Fuenfría Hospital (Fig 2D); **InP**, daily sequence of inpatients (Fig 1A); **DInP**, InP daily variation (Fig 1B); $\Xi_i$ $Rw$ (Fig 1E) built using Eq (5); **Bro**, Brownian noise calculated as indicated in the **Random fractals** Sub-subsection and Eq (7). **Whi**, White noise calculated as indicated in Sub-subsection the **Random fractals**.

$^\S$: $D_s$ is the average of $M = 100$ sequences with $N = 1329$ evaluated with Eq (14), total variance was calculated [11, 37];

$^\ddagger$, are values having borderline significance, but that could more significant if the pdf of data would have been known and the Vysochanskij–Petunin inequality use would had not been necessary, we call them 'undecided significance'.

$^\dagger$: Referred to Corollary 1 for Theorem 1.

**Statistical comparison of $D_s$ values appearing in Table 1.** $D_s$ values calculated or several white and Brown noises, as well as for La Fuenfría Hospital parameters discussed previously, are summarized in Table 1, and $P$ values to test statistical significance of differences between them appear in Table 2. An asterisk indicates marginally significant $P$ which could have been more significant if we knew $D_s$'s and the Vysochanskij–Petunin inequality would not have been necessary to test for significance [56, 57], we consider these cases of undecided significance.

In–patients sequence $D_s$ (**InP**, Tables 1 and 2, and Fig 1A) is not statistically different from from Brownian ($P > 0.13$). An asterisk indicates marginally significant $P$ which could perhaps have been more significant if the Vysochanskij–Petunin inequality use would not have been necessary [56, 57] due to the, we consider this cases as of undecided significance.

An interesting case is the $Rw$ $\Xi_i$ (Fig 1E) built for **DDiAd** (Fig 1D) using a recursive relation

$$\Xi_i = \begin{cases} 0 & \Rightarrow i = 1 \\ DDiAd_i + \Xi_{i--1} & \Rightarrow 2 \leqslant i \leqslant 1328 \end{cases} \qquad (5)$$

Please note the resemblance of sequence $\Xi_i$ (Fig 1D) and sequence in Fig 1A. Eyeball inspection of these two series shows tiny differences, yet estimates of their fractal dimension presented in Table 1 are not identical, $D_s$ for **InP** sequences is $1.33806 \pm 0.0032$ $[\overline{D}_s \pm s(\overline{D}_s)]$ calculated with Eqs (14) and (15), respectively, and is $1.33922 \pm 0.00382$ for the $\Xi_i$ sequence ($N = 1329$ in both cases).

Fig 1A and 1E are not identical, but yet their $D_s$ values are statistically indistinguishable using Vysochanskij–Petunin [See Eq (31). We have

$$\Delta D_s = D_{s,InP} - D_{s,\Xi} = 1.33922 - 1.3380 \approx 1.16 \cdot 10^{-3}$$

$$s(\Delta D_s) = \sqrt{s^2(D_{s,InP}) + s^2(D_{s,\Xi})} \approx 5.5 \cdot 10^{-3} \qquad (6)$$

$$\therefore \lambda = \frac{\Delta D_s}{s(\Delta D_s)} \approx 0.209662 < \left( \sqrt{\frac{8}{3}} = 1.63299 \ldots \right)$$

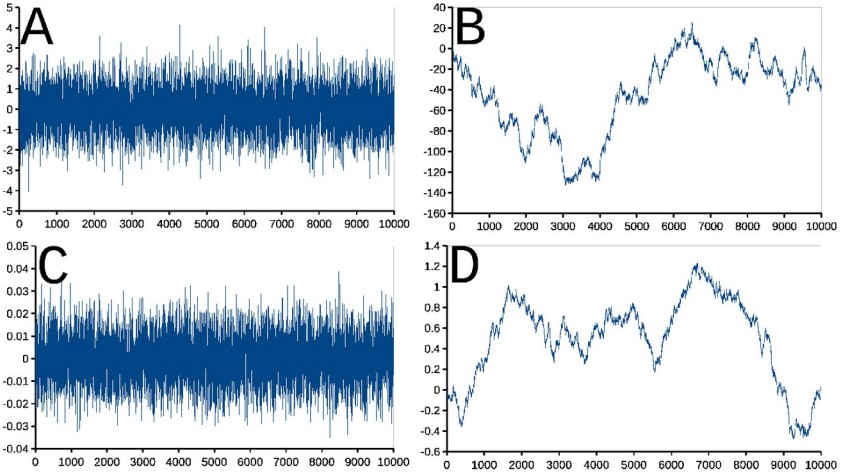

**Fig 5. Two sequences generated with Monte Carlo simulation using Eq (23) as described by Box and Muller [59, 60].** Panels are: **Panel A**, Gaussian white noise and; **B**: Gaussian $Rw$, a $Rw$ generated using Eq (7) from data on pane A; **Panel C**, Gaussian white noise and $\{g_i = N(0, 10^{-4})\}_{i=1,2,\ldots,10^4}$; **D**: Gaussian $Rw$, a $Rw$ generated using Eq (7) from data on pane C Abscissa and ordinate are arbitrary, Please notice $[D_s \pm s(D_s)]$ estimated with Eqs (14) and (15) are: Panel A, $1.66122 \pm 7.60191 \cdot 10^{-4}$; Panel B, $1.32692 \pm 7.6926 \cdot 10^{-4}$; Panel C, $1.6692 \pm 7.64176 \cdot 10^{-4}$; Panel D, $1.31723 \pm 7.72806 \cdot 10^{-4}$. $Rw$ stands for *random walk*, more details in the communication text.

which indicates that Theorem 1 does not hold to this case, and that Vysochanskij–Petunin inequality, strictly, cannot be used to asses $P(\Delta D_s)$. Yet by virtue of Corollary 1 Eq (32) indeed $P \gg \frac{1}{6} \cong 0.1\overline{6}$ for $\Delta(D_s) = 0$ occurs per chance: i.e., it is *not statistically significant* [42, 43, 58].

$\therefore$ *The **InP** Rw fluctuations were a result of delays between daily Hospital's discharges and admissions.*

**Monte Carlo simulated sequences.** White Gaussian noise in Fig 5A is a sequence of $\{g_i = N(0, 1)\}_{i=1,2,\ldots,10^4}$ and Fig 5C of $\{g_i = N(0, 10^{-4})\}_{i=1,2,\ldots,10^4}$. Brownian noise in Fig 5B is the result of

$$b_i = \begin{cases} b_1 = 0 \\ b_i + g_{i-1} \Rightarrow 2 \geqslant i \geqslant 1329 \end{cases} \tag{7}$$

where $b_i$ is the $i^{\text{th}}$ point in Fig 5B and $g_{i-1}$ is the $(i-1)^{\text{th}}$ point in Fig 5A. The required random normal variates of type $N[0, 1]$ were generated as indicated in thin the **Mathematical Appendix**. In contrast with the Gaussian white noise (Fig 5A and 5C), Brownian noise sequences have slow and fast meanders and relatively long periods with trends to increase and/or to decrease which are just local tendencies in the segment, generated as an example (Fig 5B and 5E), the segment did not indicate any periodicity. Segments such as the one in Fig 5B for $7 \cdot 10^3 \leqslant x \leqslant 10^4$ look very much like the **InP** curve at La Fuenfría Hospital in Fig 1A which also produces a false sensation of periodicity with a slight tendency to decrease, yet they are just chaotic trends, in spite of the fact that the underlying random processes (Figs 1A and 5A and 5C) have no periodicities at all. A $Rw$ expresses a system that is out of control, just like epidemics [5, 11, 44], storms [2, 61–64] or earthquakes [65], which if described with statistical parameters based on a (short) periods of time leads to wrong conclusions.

## Power spectrum analyzes

A classic form to show function periodicities in time domain, is to transform this them into frequency domain function, which is a sum of sines and cosines, using a fast Fourier transform (FFT) [39, 48]. Transformed values for each frequency ($f$) are 'complex numbers' ($\mathbb{C}$), $c_f$ comprised of two parts or components: one is sometimes referred to just the 'real component $a_f$' ($\mathfrak{R}$) and a, so called, 'imaginary' ($\mathfrak{I}$) component ($b_f\sqrt{-1}$). A complex number ($c_f$) is of the form

$$c_f = a_f + b_f\sqrt{-1} = a_f + b_f\boldsymbol{\iota} \Rightarrow \begin{cases} c_f & \in \{\mathbb{C}\} \\ a_f \wedge b_f & \in \{\mathfrak{R}\} \\ b_f\boldsymbol{\iota} & \in \{\mathfrak{I}\}, \end{cases} \tag{8}$$

where parentheses indicate sets of numbers, and $\in$ reads as "*belongs to*". The 'energy', expressed as either a steady curve's nonzero curve values, or sharp increases or variations in amplitude (such a as the peak in Fig 6B, something we could informally call its 'weight'), contributed by each frequency component to total signal energy in a fluctuating complex process

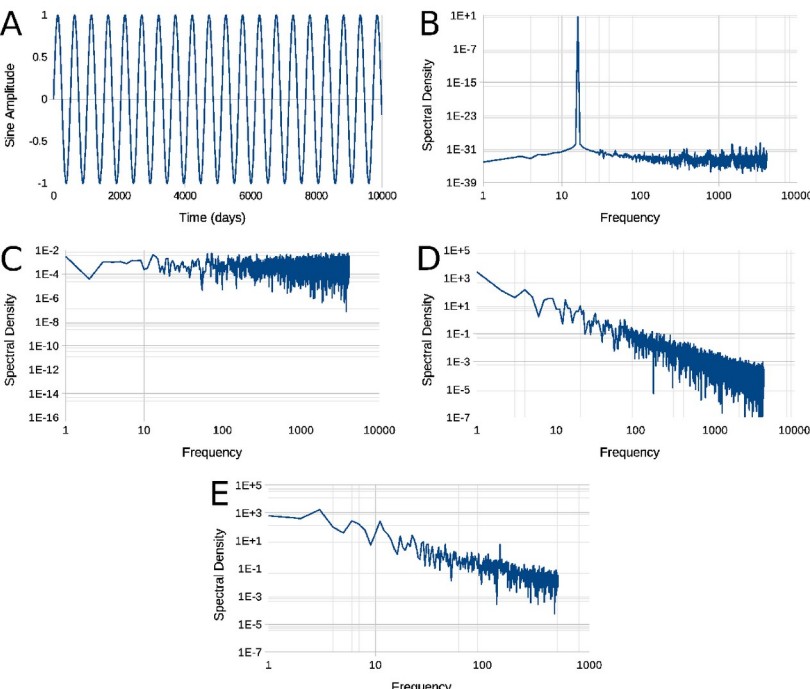

**Fig 6. False periodicity analysis in *Rw*s. Use of spectral density ($\psi_{f_j}$).** Panels are: **Panel A**- A perfectly periodic signal, the sine trigonometric function $\zeta(x) = \sin\left(x\frac{\pi}{128}\right)$; **Panel B**- $\psi_{f_j}$ of the sine function shown in panel A; **Panel C**- $\psi_{f_j}$ of Gaussian white noise shown at Fig 5A; **Panel D**- $\psi_{f_j}$ of Brown *Rw* shown in Fig 5B; **Panel E**- $\psi_{f_j}$ pf daily inpatients (***InP***) sequence which in appears in Fig 1A. Ordinate of panels B through E, is power dissipated at each frequency. The sine function depicted dissipates all its energy at a single frequency observed at a single neat peak in its $\psi_j$ (panel B). *sequence for the number of hospitalized patientsSDf_j corresponds to a Gaussian Rw without any underlying periodicity.* All SD plots were filtered using a Hann window [39, 66]. The rapid oscillations observed at right of signals in panel B to D are an artefact due to sampling discrete points in the signal for the Fourier transformation. *Rw* stands for *random walk*, further details in the text.

is given by

$$(\boldsymbol{\psi}_f = \sqrt{a_f^2 + b_f^2}) \in \mathbb{R} \tag{9}$$

plotting $\boldsymbol{\psi}_f$ against $f$ id called a *spectral density plot*, and it is areal number.

An intuitive manner to understand this, is to consider a perfect periodic signal such as a trigonometric sine function like $\zeta(x) = \sin\left(\frac{2x\pi}{T}\right)$ where $\pi = 3.1415926\ldots$ and $T$ is a constant (called the *waveform's period*) with same units (meters, seconds, grams, volts, or else) as $x$. The example is show in Fig 6A. Fig 6B, presents Fig 6A's power spectrum consisting of a single peak $f = \frac{1}{T}$ frequency. Please notice that to calculate $\boldsymbol{\psi}_f$ the actual signal is sampled in $\Delta x$ intervals (equivalent to multiplying by something called a *Dirac comb* and the FFT also contains the frequency components of the Dirac comb producing a burst of rapid vibrations in the right half of each trace [39].

Fig 6C is a $\boldsymbol{\psi}_f$ spectrum of the Gaussian white noise shown in Fig 5A; ordinates of panels B through E, in arbitrary units. In Fig 6C it is seen that all frequencies have the same 'weight', i.e., all frequencies contributions are equal. Fig 6D is a Brown $Rw$ $\boldsymbol{\psi}_f$ spectrum,; in double logarithmic coordinates, the spectral density is a straight line which decays with a slope of $\frac{1}{f^2}$. Fig 6E is $\boldsymbol{\psi}_f$ of daily inpatients (***InP***) sequence shown in Fig 1A.

Although the simulated sequences in Fig 6 are 10000 points long and Hospital data are only 1329 points long, there is an evident similarity between the Brownian noise $\boldsymbol{\psi}_f$ (Fig 6D), and $\boldsymbol{\psi}_f$ from the sequence of Hospital inpatients (***Inp***) between May 1 2013 and December 19 2017 (Fig 6E). In both cases $\boldsymbol{\psi}_f$ are characteristic of a Brown $Rw$ and none of these $\boldsymbol{\psi}_f$s has a peak which could indicate a predominant periodic component which could suggest for any periodicity, despite of short lapses in the Brown $Rw$s that could give this impression.

Fig 6C is Gaussian white noise $\psi_{f_j}$ calculated for the signal in Fig 5A. In Fig 6C all frequencies have the same 'weights', the contributions of al frequencies are equal, which leads to the 'white' name of this kind of noise. Fig 6D shows $\psi_{f_j}$ of the Brownian $Rw$ in Fig 5B, one may observe that in double logarithmic coordinates the spectral density follows a straight line which decays with a slope of $\frac{1}{f^2}$. Finally, also in Fig 6E, we show $\psi_{f_j}$ corresponding to daily hospitalized patients sequence shown in Fig 1A. Except for that simulated sequences in Fig 6 are $10^4$ points long and Hospital data are only 1329 points long, there is an evident similarity between the Brownian noise $\psi_{f_j}$, and that $\psi_{f_j}$ from the sequence of Hospital discharges between May 1 2013 and December 19 2017. In both cases $\psi_{f_j}$ are characteristic of a $Rw$ and *none of these $\psi_{f_j}$ has no peak which could suggest periodic components which could account for any periodicity, despite short periods in the Rws that could give this impression.*

## Daily fluctuations in stay duration and average stay duration in operative units at La Fuenfría Hospital

The Hospital is divided into the following Clinical Operative Units (**COU**): **RCPU**, Relapsing Chronic Patients Unit, this unit operated until May 2016 when its beds were transferred to the **CCU**; Chronic Care Unit; **PCU**, Palliative Care Unit; **FRU**, Function Recovery Unit; **TU**, Tuberculosis Unit and **NTU**, Neurorehabilitation Treatment Unit. Fig 7A presents data on daily number of in–patients (InP) at each unit, and their average stay duration on Panel Fig 7B). Daily fluctuations of in–patients and average length of stays (**LoS**) for each COU calculated every day from January 1 2015 uo to December 12 201. Abscissas are days from January 1 2014, ordinates are in decimal logarithmic scale to better visualize some of sequence large

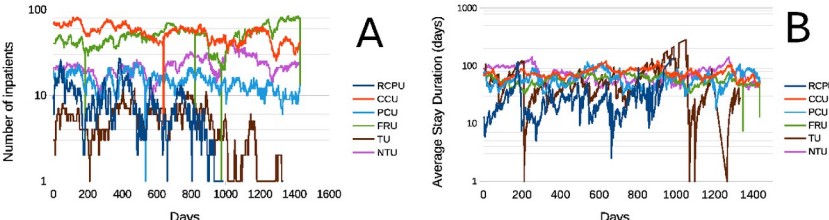

**Fig 7. Daily fluctuations of in-patients and average duration of stays for each Clinical Operative Unit recalculated every day from January 1 2015 until December 12 2017. A-** Number of in–patients (**InP**) at each Operative Unit. **B-** Average stay duration at each Operating Unit [$Y_d$ defined as in Eq (10)]. Abscissas are days elapsed from January 1 2014, ordinates are in logarithmic scale to better visualize some of sequence features. Unit initials mean: **RCPU**, Relapsing Chronic Patients Unit, this unit was suppressed on May 2016 and its beds were transferred to **CCU**; **CCU**, Chronic Care Unit; **PCU**, Palliative Care Unit; **FRU**, Function Recovery Unit; **TU**, Tuberculosis Unit; **NTU**, Neurrehabilitation Treatment Unit.

outliers, which are particularly evident in Panel Fig 7B. Traces in Fig 7 seem to be a kind of *Rw*. All have some outlying peaks, particularly RCPU and UTB, however, less prominent peaks, occur in other COU too. It it is important to note that in this paper we sometimes refer to products or quotients between random variables, this is the case of averages including number of patients and stay duration in each COU in Fig 7B. We know that our data are not Gaussian, however even in the case of Gaussian values these averages, may be quotients of random variables following a a "pathological" statistical distribution such as the Cauchy pdf (See the Mathematical Appendix **Subsection On Cauchy distributed variables**).

## Nonparametric statistical analysis of Hospital's operative units sequences shown in Fig 7

"If a sample is too small you can't show that anything is statistically significant,

whereas if sample is too big, everything is statistically significant.". . . "you are doomed to significance."

Norman and Steiner [72, pg. 24].

As said in Section sequences in Fig 7B include data such **InP** average patients' stay duration in each operative unit (**Y**). This number calculated as:

$$\overline{Y} = \frac{\sum_{j=1}^{N_d} Y_{d,j}}{N_d} \tag{10}$$

where $d$ represent a given period and $j$ is the $j^t$h patient in the service, and $N_d$ is the number of patients in each service at day $d$. Thus, $Y_d$ are quotients unknown distribution random variables. As mentioned in the **Mathematical Appendix**, those quotients of random variables have uncertain distributions, which in the case of Gaussian variables is the Cauchy pdf Eq (24) discussed in the **Mathematical Appendix**. These quotients follow unknown pdfs when terms in the ratio obey unknown pdfs. For these distributions sample mean ($\overline{x}$) n and sample variance [$s^2(x)$] are meaningless. Statistical sampling theory is based on the fact that $\overline{x}$ and $s^2(x)$ converge towards the population mean ($\mu$), and population variance [$\sigma^2$]. Yet, $\mu$ and $\sigma^2$ do not exist for Cauchy and other, so called, "pathological distributions" for which $\overline{x}$ and $s^2(x)$ have nowhere towards to converge to. Cauchy sample experimental estimates of $\overline{x}$ and $s^2(x)$ vary

**Table 3. Nonparametric linear Theil [67–70] regression parameters of LFH clinical units' time sequences expressed for each unit as in Fig 7.**

| Sequences | $\alpha$ | $100 \cdot \beta$ | $\rho_S$ | P | N |
|---|---|---|---|---|---|
| **InP** | | | | | |
| RCPU | 12.8 (12.5, 13.0) | −1.0 (−1.1, −0.9) | −0.618 | $<10^{-6}$ | 986 |
| CCU | 67.9 (67.4, 68.4) | −2.9 (−2.1, −1.9) | −0.695 | $<10^{-6}$ | 1434 |
| RCPU | 18.3 (18.1, 18.4) | −0.5 (−0.6, −0.5) | −0.540 | $<10^{-6}$ | 1434 |
| RFU | 37.7 (37.1, 38.2) | 1.9 (1.8, 2.1) | 0.558 | $<10^{-6}$ | 1434 |
| TB | 6.6 (6.5, 6.8) | −0.3 (−0.33, −0.26) | −0.447 | $<10^{-6}$ | 1331 |
| TNU | 18.8 (18.5, 19.1) | 0.5 (0.4, 0.6) | 0.4231 | $<10^{-6}$ | 1434 |
| $\overline{Y_d}$ | | | | | |
| RCPU | 16.1 (15.2, 16.9) | 1.8 (1.5, 2.1) | 0.343 | $<10^{-6}$ | 985 |
| CCU | 67.9 (67.4, 68.4) | 0.1 (−0.09, 0.30) | 0.002 | 0.96 | 1434 |
| CPU | 54.4 (53.5, 55.4) | 0.6 (0.4, 0.9) | 0.123 | $2 \cdot 10^{-6}$ | 1434 |
| RFU | 59.6 (59.0, 60.2) | −0.3 (−0.5, −0.2) | −0.107 | $4 \cdot 10^{-5}$ | 1434 |
| TB | 53.8 (51.6, 56.1) | −0.9 (−1.3, −0.4) | −0.181 | $<10^{-6}$ | 1331 |
| TNU | 84.1 (83.0, 85.3) | −1.7 (−2.0, −1.4) | −0.312 | $<10^{-6}$ | 1434 |

Sequence of **InP**: number of in-patients in the Unit. $\overline{Y_d}$ sequences are average stay duration at each Operating Unit [defined as in Fig 7 by Eq (10)]. Data presented as medians and their 95% Confidence Interval (in parentheses), $\alpha$ regression line value when $x = 0$ (intercept), $100 \cdot \beta$ straight line regression coefficient (slope, multiplied by 100 to reduce Table width), $\rho_S$ nonparametric Spearman's [71] crenelation coefficient, P is the probability that each sequence comes from a population with $\rho_S = 0$, usually called the probability that the correlation is not statistically significant, and N is the number of data pairs used for the linear regression analysis. HLF unit initials mean: **RCPU**, Relapsing Chronic Patients Unit, this unit was suppressed on May 2016 and its beds were transferred to **CCU**; **CCU**, Chronic Care Unit; **PCU**, Palliative Care Unit; **FRU**, Function Recovery Unit; **TU**, Tuberculosis Unit; **NTU**, Neurorehabilitation Treatment Unit.

wildly, specially when sample sizes grow and large outlier values become very likely [73, 74]. Due to these only nonparametric descriptive analysis provides meaningful information on sample localization and dispersion.

*Caveat emptor*: data samples from La Fuenfría Hospital, except for those in Fig 3, are rather large ($986 \leqslant N \leqslant 1434$). Under these circumstances small, but statistically significant changes, which lack relevance or meaning in practice, may be detected, [70, 75, 76]. Table 3 contains results where sequences in Fig 3 are subjects o a nonparametric Theil regression analysis [77]. As seen in Table 3, many of the hospital units sequences in Fig 6 seem to produce a small, but highly statistically significant, correlation between the sequence under study and time. All the time series in this figures are *Rw*s (not shown), as was confirmed by their spectral density, characteristic of Brownian noise, without any evidence of periodicity, and resembling those in Fig 7D and 7E. As stated, in connection with the sequence in Fig 5B and 5D, *Rw*s may some times exhibit rather long, trends to increase or decrease, that are independent from any variation in factors determining the *Rw*. The pdf of $\Delta y_t = y_t - y_{t-\Delta t}$ did not change in those periods. Due to these meanderings of time sequences, descriptive statistical parameters, even if optimally chosen, like in Table 3 fail to provide meaningful information on *Rw*s like Fig 6B and some of the preceding figures.

## Discussion and conclusions

### Study purpose an empirical finding

At least during the study period, the La Fuenfría Hospital operated, in median, at 76.8% of its total capacity and fluctuated in what could be interpreted as slow periodic meanders (Fig 1A). Yet, the daily admissions (Fig 1B) or discharges from LFH (Fig 1D) are random sequences which show no periodicities whatsoever. Daily differences (Fig 1D) between patients admitted

(Fig 2B) and discharged (Fig 2D) from the Hospital were also random non periodic sequences. It was somewhat surprising to find that the operating Hospital's curve (Fig 1A)) could be almost perfectly (Fig 1E) reproduced when daily differences between admitted and discharged patients (Fig 1D) were added sequentially as indicated in the **Random fractals** Section's Eq (3).

### Meaning of *lack of control* of a process

Data on Figs 1A and 6E reflect that Hospital bed occupancy is described by a random fractal function, the so called Brownian *Rw*. Just like Brownian *Rw*, bed occupancy fluctuates with important periods of low occupancy.

The main characteristic of Brownian *Rw*. is that its complexity (reflected in its $D_s$) does not change if the number of daily inpatients (**DInP**) increases or decreases, this is called self affinity of the process [5, 44], it does not change with scaling **DInP**. This is shown by the Monte Carlo simulated traces of Fig 5 where the Gaussian white noise $\{g_i = N(0, 1)\}_{i=1,2,...,10^4}$ in panel Fig 5A looks very similar to the $\{g_i = N(0, 10^{-4})\}_{i=1,2,...,10^4}$ white noise in panel Fig 5C, exp of course for the ordinate scale; Something similar happens with the associated Brownian noises in panels Fig 5B and 5D, which have similar complexities, but the fluctuations in Panel Fig 5B are approximately 10 times wider in than in Panel Fig 5D.

*Still, from a production efficiency view point if the differences (in Fig 5 ) are very important since the fluctuation range of the Brownian Rw diminishes when $g_i \rightarrow 0$ and production of goods or services increases as the system becomes more predictable (controlled), uniformly productive and efficient.*

At La Fuenfría Hospital **InP** would fluctuate less if the lag between **Adm** and **Dis** $\rightarrow 0$. Identifying a problem is the first step to be able to solve it. This means, making the Hospital more efficient requires analyzing the factors that keep **Adm** and **Dis** out of synchrony.

### Characteristics of La Fuenfría Hospital [78]

Hospital La Fuenfría (https://bit.ly/32FVmyf) is one of three medium and long stay SERMAS hospitals in Madrid's Autonomous Region (MAC) together with Virgen de la Poveda Hospital and Guadarrama Hospital, under the Sanitary Chancellery (*Consejería de Sanidad*), LFH is in Cercedilla Municipality at the north of Madrid Region, at 60 km from Madrid city. The MAC comprises an area of of 8021.8 km$^2$ and a population of 6.6 M.

The long stay duration are related to local norms and procedures to handle patients, it is likely that in Spain the public health system has more leeway to tolerate longer stays than perhaps other EU nations, especially when dealing with needy patients. Concurrently, LFH manages many chronic cases, and it most important unit is FRU, responsible for recovering patients with disabilities stemming from bone surgery (prothesis, fractures, etc.) and motor deficiencies following cerebrovascular accidents (CVA or strokes). Thus perhaps it is to be expected that stays at LFHa may be longer than at hospitals dealing under ambulatory treatment.

LFH is located in a pleasant rural area, Madrid's Sierra de Guadarrama, convenient for its goals, but distant from Madrid (population 3.3 M) city and MAC, in general. Patients are referred from other SERMAS hospitals, thus, delays between LFH discharges and new patients arrivals, are not surprising. The analyst of data in this paper (CS) was completely unaware of this factors, but they popped out sharply from data analysis discussed here.

### Classical statistical analysis of institutional efficiency

Descriptive statistics, using parametric (Gaussian) methods, is probably the the most common manner used tools to gauge institutional efficiency. LFH SELENE$^{©}$ include command panels

parameters that are ratios of random variables. These ratios have unknown statistical distributions which like, the Cauchy pdf, have no central moments, therefore mean or variances do not exist, thus textbook recipes like

$$\overline{x} = \frac{\sum_{i=1}^{N} x_i}{N} \text{ and/or } s^2(x) = \frac{\sum_{i=1}^{N} (x_i - \overline{x})^2}{N - 1},$$

for sample mean and variance, are, calculable, yet, absolutely meaningless [79–82] (See the section on the Cauchy distribution in the **Mathematical Appendix**. For these kinds of random samples the use of nonparametric statistical estimators such as medians and 95% CI must always be preferred to $\overline{x}$ and $s^2(\overline{x})$ which, are also, very sensitive to outliers and, as said, are meaningless for Cauychyan variables. Furthermore, in the analysis of data extracted from LFH command panels, no Gaussaian variables were found, further demanding the use of nonparametric statistical methods.

Another example of the failure of statistics to productively help analyze aspects of LFH operative efficiency, may be derived from Fig 7. The figure presents unit operation data are *Rw*s with large outliers. However, when these *Rw* were subjected to linear nonparametric regressions (Table 3) apparent correlations with time appeared, with correlation coefficients statistically distinct from zero, i.e., apparently *highly* unlikely to differ from zero per chance. Large samples are a good way to discover statistically significant, but irrelevant, trends, which, as shown here, may just be temporary tendencies of a random walk. Such correlations could be used to create institutional (probably wrong) operation policies. Classical statistics is notoriously limited to study fractal processes [4, 5, 7, 35, 36, 44]. Therefor, of our findings regarding statistical evaluation of LFH operational curves are neither new nor surprising. The novelty of our findings relates to the commonality of statistical analyses and the lack of other analyzes (fractal or else) to study institutional efficiency.

Several caveats should be considered:

1. This communication is centered on maximizing the use of hospital facilities as an indication of efficiency. Yet, it does not consider other criteria such as the patient's welfare, degree of recovery, speed of recovery, which are of fundamental importance, but also depend on the efficient of hospital resources.

2. The main theoretical tool we use is fractal analysis, but to accurately estimate fractal dimension very long data series [11, 37, 83]. this limitation, as done here, may be overcome py the use of sequences with known fractal dimensions as applied here and in previous studies [11, 37].

3. Spectral analysis confirmed the lack of periodicity in the **InP** time series.

4. Yet, the most important finding reported here is that that the wide fluctuations observed in the inpatients (**InP**, Fig 1A) may be reconstructed with identical naked-eye and fractal properties ($\Xi_i$ sequence in Fig 1E) by adding the daily differences in LFH patients admitted and discharged (**DDiAd**, Fig 1D) using Eq (5). *These are empirical finding result from any mathematical theory or artifact*.

### *Rw* in La Fuenfría Hospital operation curves

The most surprising finding in this study, is that curves representing in-patients, **InP**, and **ΔInP** are random Markovian processes [51]. In a Markovian processes the past determines what happens now, and what occurs now decides the future. If there were no random

components in the sequence, known the present would allow to know past and future of the system. Yet, since uncertainty separates one instant from the next, the whole system becomes unpredictable. It dt drifts over time in ways that a naive observer can realize, one is driven to believe that the system is predictable and determined.

If a present drift trend pleases us, we tend to believe that whatever we are doing is "correct", in the opposite case we would tend to believe that we are doing something "wrong", even if in reality we have no "merit" in the first case nor are we "guilty" in the second case. An examples of this is seen at times from 0 to 6000 in Fig 5B. But after that time the *Rw* in Fig 5B which produces a false appearance of relative "stability", even when at all those times it is exactly the same system under the same $\{g_i = N(0, 1)\}_{i=1,2,...,10^4}$ Gaussian conditions as shown by the plot in Fig 5A.

## Caveats and recommendations

### The contribution of our work

We know (both authors are trained MDs) that hospitals and medicine are complex systems, yet our study shows that the waveform meandering between full and insufficient occupancy at LFH may be perfectly, explained considering a single factor: admission and discharges are systematically and randomly out of phase. These random differences, small as they may be, constitute the de determining factor at LFH: To the best of our knowledge nobody has previously singled this small but important hidden factor. We know (both authors are trained Mds) that hospitals and medicine are complex systems, yet our study shows that the waveform meandering between full and insufficient occupancy at LFH may be, exactly, explained considering *a single factor: admission and discharges are systematically and randomly out of phase*. This random difference, is the determining factor at LFH: To the best of our knowledge nobody has previously singled out this small but important hidden factor.

### Detecting problem indicators

Descriptive statistics and common *Command panels*, are not enough to detect hidden administration problems with impact on hospital efficiency. Descriptive statistics limitations mentioned in in the prior Subsection **Rw in La Fuenfría Hospital** operation curves open a question on how to evaluate and optimize the operation and administration of La Fuenfría Hospital whose operation curves are Markovian random walks.

Random walk curve meanders reduce system efficiency. In a hospital performance case, low occupation equals inefficient periods, for the following reasons:

1. The population demanding service, does not receive such service..

2. The cost of operating the hospital operation costs (infrastructure, salaries, etc.) are wasted during periods of low occupancy

At la Fuenfría Hospital occupation was as low as 45% (Fig 1A) of its capacity. As discussed in the **La Fuenfría Hospital data statistical characteristics** Sub-section oscillations between full and partial occupation seem related to a daily lag between Admissions and Discharges. If this lag is eliminated, fluctuations in occupancy would disappear should this lag be eliminated fluctuations in occupancy would disappear.

During the study period, the mean number of LFH in-patients was $\overline{InP} = 149.2$ almost identical to its median $\widehat{InP} = 150.0$, and ranged (102, 187). Since the hospital had 192 patient's beds, thus the bed occupancy during the study period was $\overline{x} = 77.7\%$ and a median of $\hat{x} = 78.2$ ranging (53.1, 97.4)%. Thus over the observation period, on average 22% ranging

(3, 47)% La Fuendría Hospital beds were vacant, which means that a similar proportion of the resources assigned to LFH was lost, and therefore this does not generate a lack of demand.

## Can the problem be solved?

Our study indicates that LFH vacancies could be eliminated or greatly reduced if the daily difference **Adm-Dis** is nullified (see **Meaning of *lack of control*** Sub-section) of a process, Figs 2 and 5). Efficient service characterized by a *Rw* in time which may be increased by reducing broad meanders of the walk (Fig 5). In the case of La Fuenfría Hospital this study suggests that the key factor for reducing efficiency is the daily difference between admissions and discharges.

Nonetheless, the solution probably transcends the realms of La Fuenfría Hospital. We have no information on other SERMAS hospitals, yet the description, La Fuenfría Hospital realms. We have no information on other SERMAS hospitals, yet the description of LFH provided in **Introduction** Section illustrates that LFH is interwoven in the SERMAS network, seems necessary to consider current communications and required modifications within the network. The easy aspect may be to improve communications between admission and discharge departments within LFH, although this may imply some changes to the internal computer network. Thus, if the admissions department were to be given notice prior to the actual discharges, this would enable the possibility to notify the SERMAS network that LFH will have space for a new patient and ensure that the hospital is ready to occupy the space immediately when it actually becomes available.

However, this immediate availability is also hindered by the large area that SERMAS covers, the fact that there is a distance of 60 km from the city of Madrid (the largest set of SERMAS patients), and even more from other areas in Madrid's Autonomous Region. Factors such as means and speed of transportation may be critical, institutional transportation within the SERMAS network is probably faster and easier to program, than solely relying on the patient's and its family's resources and efficiency, for example. However, if the efficiency of SERMAS hospitals is affected to the same extent as La Fuenfría Hospital's, it may be worthwhile to study the system and to implement solutions between admission and discharges in SERMAS hospitals to improve the system's efficiency.

## Appendices

### Mathematical appendix

**Calculating *D*, approximated fractal dimension of sequences.**  Classical (Euclidean) geometry is useful to describe relatively simple forms such as straight lines, squares, circles, cones, pyramids, etc. All these forms may be represented as a linear combination of an integer number of orthogonal components called Euclidean dimensions. One- or two-dimensional Euclidean spaces are subsets of our sensory tri-dimensional space. Yet, almost all natural forms cannot be represented in terms of Euclidean geometry.

According to Mandelbrot [5, pg. 15 and Ch. 39]:

"A fractal is by definition a set for which the Hausdorff– Besicovitch dimension strictly exceeds the topological dimension. Every set with a non-integer *D* is a fractal."

The Hausdorff–Besicovitch dimension of a set in a metric space may be expressed as [5]:

$$D_h = -\lim_{\epsilon \to 0} \frac{\ln[N(\epsilon)]}{\ln(\epsilon)} = -\lim_{\epsilon \to 0} \log_\epsilon[N(\epsilon)] \tag{11}$$

where $N(\epsilon)$ is the number of open balls or radius $\epsilon$ required to cover the set. In a metric space given any point $X$, an open ball centered on $X$ with radius $\epsilon$, is the set of all points $x$ for which distance between $X$ and $x$ is $< \epsilon$.

The term waveform is applied to the shape of a set, usually drawn as instant values of a periodic quantity in time. Besides classical methods such as moment statistics and regression analysis, properties like Kolmogorov y Sinai entropy [84], apparent entropy [7, 85, 86] and fractal [11] have been proposed to analyze waveforms. Fractal analysis may provide information on spacial extent (tortuosity or capacity to extend in space) and on self-similarity (the property of staying unchanged when measure scale changes) and self affinity [87]. In bidimensional spaces waveforms are planar curves, which may have coordinates of different units or dimensions.

There is a simple algorithm to approximate fractal dimension of a curve [11]. To achieve this, fractal dimension $D_s$ is estimated for a set of $N$ points from a set $\{x_i, y_i\}_{i=0,1,2,\dots,N}$ values were the abscissa $x_i$ increases by a constant factor ($\Delta x$). The curve is transformed into a unit square (a square with side length equal to 1) [11] as follows. A first transformation normalizes every abscissa point as:

$$x_i^* = \frac{x_i}{x_{max} - x_{min}} \tag{12}$$

where $x_i$s are the original abscissa values, $x_{max}$ is the largest $x_i$. A second transform normalizes the ordinate as:

$$y_i^* = \frac{y_i - y_{min}}{y_{max} - y_{min}} \tag{13}$$

where $y_i$ are the original ordinate values, and $y_{min}$ and $y_{max}$ are minimum and maximum $y_i$, respectively. The unit square may be seen as covered with a $N \cdot N$ grid of cells. $N$ containing a transformed curve point. A length $L$ line may be divided into $N(\epsilon) = L/(2 \cdot \epsilon)$ segments of mean length $2 \cdot \epsilon$, and covered with $N$ radius $\epsilon$ open balls. Fractal dimension is the approximated as $D_s$ [11, Ecuation (6a)] as

$$D_h = \lim_{N' \to \infty} \left[ D_s = 1 + \frac{\ln(L) - \ln(2)}{\ln(2 \cdot N')} \right] \tag{14}$$

where $D_h$ is again the Hausdorff–Besicovitch dimension, $N' = N - 1$, and $L$ is the curve length after embedding in the unit square. *It is important to observe that $D_s = D_h$ only at the limit when $N' \to \infty$, for all other $N'$ values, $D_h > D_s$.* Thus $D_s$s obtained with same $N'$s are the only fit to be compared. $D_s$ uncertainty may be estimated through its variance [11]:

$$
\begin{aligned}
\text{var}(D_s) \quad &= \frac{N'}{L^2 \cdot \ln(2 \cdot N')^2} \sum_{i=1}^{N} \frac{(\Delta y_i - \overline{\Delta y})^2}{N'} = s^2(D_s) \\
&\Rightarrow s(D_s) = \sqrt{\frac{N'}{L^2 \cdot \ln(2 \cdot N')^2} \sum_{i=1}^{N} \frac{(\Delta y_i - \overline{\Delta y})^2}{N'}}
\end{aligned}
\tag{15}
$$

where $\Delta y_i$ is segment $i$ length, distance between points $(x_{i-1}, y_{i-1})$ and $(x_i, y_i)$ of the transform and $\overline{\Delta y}$ is those segments' mean. $D_s$ means and variances may be used to compare empiric $D_s$ values [35, Sections 5.2.1 and 5.2.4] [37] based on Vysochanskij–Petunin inequality [56, 57].

**Generation of random sequences for calibration purposes.** Monte Carlo simulation was used to generate $M = 100$ traces of $N = 1329$ points each of white noise (see **Random Fractals** Subsection) of *white noise*. White noises simulated were sequences of Gaussian variables of type $N(0, 1)$ [Eq (23)]. Estimated fractal dimension $D_s$ was calculated for each simulated trace

as indicated in Subsection, $D_s$ [Eq (14)] and var($D_s$) [Eq (15)]. Eq (15) intra sequence $D_s$ variance, due to unit square–embedded waveform $\Delta y_i$ in each $j$th sequence. Each sequence is a (short?) segment from an infinite sequence with same properties. Every sequence of length $N < \infty$ generated as indicated may be used to produce a different approximation to $D_h$ depending on where each is sample taken. To estimate $D_s$ and its uncertainty means were calculated for $M$ simulated traces as

$$\overline{D}_s = \sum_{j=1}^{M} \frac{D_{s,j}}{M}. \tag{16}$$

The variance od $D_s$ **between** curves (which may be visualized as random sets $N$ points sampled from an infinitely curve) for the $M$ waveforms sets was determined as

$$\mathrm{var}_b(\overline{D}_s) = \sum_{j=1}^{M} \frac{\left(D_{s,j} - \overline{D}_{s,w}\right)^2}{M-1}. \tag{17}$$

Then, total variance, $\mathrm{var}_t(\overline{D}_{s,w})$, was determined as

$$\mathrm{var}_t(\overline{D}_{s,w}) = \mathrm{var}_b(\overline{D}_s) + \left[ \overline{\mathrm{var}_w(D_s)} = \sum_{j=1}^{M} \frac{\mathrm{var}(D_{s,j})}{M} \right], \tag{18}$$

where $\overline{\mathrm{var}_w(D_s)}$ is the average variance within traces and var($D_{s,j}$) calculated with Eq (15).

$\overline{D}_{s,b}$ and $\mathrm{var}_t(\overline{D}_s)$ were used to compare the difference in $D_s$ from the empiric sequences with Monte Carlo simulated sequences with known properties. In this manner it was possible to decide if empiric sequences were white, brow, or other kind noise, spite of $N < \infty$ [11, 37]. Statistical significance of the differences was established using the Vysochanskij and Petunin inequality (V-P, details in Theorem 1). V-P inequality allows comparing means of samples coming from a population with *unknown pdf as long as it is unimodal, and has a definite variance* [56, 57, 88]. Due to the unknown pdf, the comparison using V-P inequality tends to declare as not different to means that are weakly statistically significantly distinct (statistical error of type II [28]), but is $\left(1 - \frac{4}{9}\right) \cdot 100 \approx 56\%$ less likelier to produce type II errors than classical Tchebichev inequality [89], of which V-P inequality is a particular case. This means that parameter differences declared significant with the V-P inequality, are more significant than they appear. To compare the statistical significance of a difference between two $D_s$ means [35, Adaptwd from Eqs. (34) and (35)] it is required only that

$$P\big(|\overline{D}_{s,1} - \overline{D}_{s,2}| = 0\big) = P\Big(|\Delta \overline{D}_{s(1,2)}| = 0\Big) \leq \frac{2}{9\sqrt{\mathrm{var}_t(\overline{D}_{s,1}) + \mathrm{var}_t(\overline{D}_{s,2})}} \tag{19}$$

where the subindex t indicates total variance, and

$$P\left[ \frac{\Delta \overline{D}_{s\,1,2}}{\sqrt{\mathrm{var}_t(\overline{D}_{s_1}) + \mathrm{var}_t(\overline{D}_{s,2})}} \geq \left( \xi = \sqrt{\frac{40}{9}} = 2.108\ldots \right) \right] \leq 0.05 = \alpha \tag{20}$$

where $\alpha$ is the largest probability we accept to declare a difference statistically significant. If $D$'s pdf is known and Gaussian Eq (20) constant would not be $\xi = \sqrt{\frac{40}{9}} = 2.108\ldots$ as specified by the V-P inequality, but $\xi = 1.959\ldots$. This is true for any case where the pdf is known, thus we

should write Eq (20) somewhat like $\xi \lessgtr \sqrt{\frac{40}{9}}$ within the parentheses, but this is unnecessary due to the demonstrated inequality [56, 57, 88].

Eqs (16) to (18) were also calculated for sets of $M = 100$ Brownian sequences with $N = 1329$ like

$$b(t + \Delta t) = b(t) + N(0,\ 1) \tag{21}$$

and were used as standards to calculate the probability of significance between their estimated $\overline{D}_s$ versus $D_s$ calculates for and empirical distribution of interest. Their fractal dimension $D_h$, when $N \to \infty$ is 1.5. An example may be found in [37]. The $N[0, 1]$ variable was generated as indicated in Eq (23).

**Generating uniform (rectangular) random variables.** Uniform random variables of $U[0, 1]$ type were generated. Fundamental to all Monte Carlo simulations [90] is a good uniform (pseudo) random (PRNG) number generator. Data for all numerical simulations carried out in this work were produced using random numbers ($r$) with continuous rectangular (uniform) distribution in the closed interval [0, 1] or $U[0, 1]$. All $U[0, 1]$ [Eq (22)] were generated using the 2002/2/10 initialization-improved 623-dimensionally evenly distributed [91] uniform pseudo random number generator MT19937 algorithm [92, 93]. The procedure used makes exceedingly unlikely ($P = 2^{-64} \approx 5.4 \cdot 10^{-20}$) that the same sequence, $\{r_i\}$, of $U[0, 1]$ is used twice. Calculations were programmed in C++ using g++ version 5.4.0 20160609 with C++14 standards, under Ubuntu GNU Linux version 18.4. For further details on random number generator initializing see Sevcik [37].

Rectangular random variables fulfill the following property

$$U[a, b] = \begin{cases} 0 & \Rightarrow x < a \\ k = \dfrac{1}{b - a} & \Rightarrow a \leqslant x \leqslant b \\ 0 & \Rightarrow x > b \end{cases} \tag{22}$$

with mean $E(x) = \frac{b-a}{2}$, $x$, $k$ may be a real number [11] or an integer [11, 37].

**Generating random normal deviates.** Random normal variables of the ding $N[\mu, \sigma] = N[0, 1]$ were generated with the Box and Muller [59, 94] algorithm

$$N[0, 1] = \begin{cases} \sin[2\pi r_1]\sqrt{-2\ln[r_2]} \\ \\ \cos[2\pi r_1]\sqrt{-2\ln[r_2]} \end{cases} \tag{23}$$

where $r_1$ and $r_2$ are two uniformly distributed random variates of the kind $U[0, 1.]$.

**On Cauchy distributed variables.** The Cauchy distribution is symmetric about its median (location factor, $\hat{\mu}$) and has a width (dispersion) factor ($\lambda$) and its pdf is

$$c(x) = \frac{1}{\pi} \cdot \frac{\lambda}{\lambda^2 + (x - \hat{\mu})^2} \tag{24}$$

and also has a PDF like

$$C(x) = \frac{1}{\pi} \cdot \arctan\left(\frac{x - \hat{\mu}}{\lambda}\right) + \frac{1}{2} \tag{25}$$

where *arctan* refers to the *arc tangent* trigonometric function. Cauchy pdf has no central

moments [28, 81], and thus has no defined mean, variance, skewedness or kurtosis [75, 81, 95]. If conventional equations for sample $\overline{y}$ or $s^2(y)$ values are calculated, they will be seen to vary wildly and wild variability increases with sample size. Cauchy–type data can be described and compared only with nonparametric statistics. From Eq (25) it is possible to prove that a 95% CI of a Cauchyan variable is equivalent to $\approx \mu \pm 12.706\lambda$ which gives a $\approx 25.412\lambda$ span to the 95% CI. A Cauchy random variable is prone to assume values very far away from its median, outliers are very commonly observed as sample size increases [75, 81, 95].

**The Vysochanskij-Petunin inequality [56, 57].** **Theorem 1** (Vysochanskij-Petunin). : *Let X be a random variable with unimodal distribution, mean μ and finite, non-zero variance $\sigma^2$. Then, for any $\lambda > \sqrt{\frac{8}{3}} = 1.63299\ldots$*

$$P(|X - \mu| \geq \lambda\sigma) \leq \frac{4}{9\lambda^2} = \epsilon. \tag{26}$$

## Using the inequality to calculate significance when $\lambda > \sqrt{\frac{8}{3}}$

Theorem 1 holds even with heavily skewed distributions and puts bounds on how much of the data is, or is not "in the middle". Setting $\epsilon = 0.05$ then $\lambda = \pm\left(\sqrt{\frac{80}{9}} \approx 2.981\right)$, for a two tailed test. By virtue of Eq (26) no matter which unimodal distribution, no matter how skewed, there will be <2.5% chance that a datum will belong to a population with mean $\mu$ and variance $\sigma^2$ if it lays farther than $\pm2.981\sigma$ from $\mu$. Please note that if the probability distribution function of data is Gaussian $\pm1.96\sigma$ suffices to reach the same confidence level. We introduced the $\epsilon$ variable in Eq (26) for the statistical argumentation that follows.

Let $\overline{D}_{s,1}$ and $s^2(D_{s,1})$ be the mean and variance estimated for the random variable $D_{s,1}$, and $\overline{D}_{s,2}$ and $s^2(D_{s,2})$ be the mean and variance estimated for the random variable $D_{s,2}$ (linearly independent from $D_{s,1}$), then

$$\Delta\overline{D}_{s,(1,2)} = \overline{D}_{s,1} - \overline{D}_{s,2} \tag{27}$$

and the estimated variance of $\Delta\overline{D}_{s,(1,2)}$ is

$$s^2(\Delta\overline{D}_{s,(1,2)}) = s^2(D_{s,1}) + s^2(D_{s,2}). \tag{28}$$

By virtue of the V-P inequality [Eq (26)]

$$P\left(\frac{|\Delta\overline{D}_{s,(1,2)}|}{s(\Delta\overline{D}_{s,(1,2)})} \geqslant \sqrt{\frac{4}{9\lambda^2}}\right) = P\left(\frac{2|\Delta\overline{D}_{s,1,2}|}{3s(\Delta\overline{D}_{s,(1,2)})} \leqslant \lambda\right) \leqslant \epsilon \tag{29}$$

which means Eq (26) that

$$\epsilon = \frac{2}{3\left[\dfrac{\Delta\overline{D}_{s,(1,2)}}{s(\Delta\overline{D}_{s,(1,2)})}\right]}. \tag{30}$$

Therefore, there is $\leq \epsilon$ probability that $|\Delta \overline{D}_{s,(1,2)}| \neq 0$ due to random sampling variation, and $\Delta \overline{D}_{s,(1,2)} \neq 0$ with a confidence level $P \leq \epsilon$. Thus:

$$P\left[\frac{|\Delta \overline{D}_{s,(1,2)}|}{s(\Delta \overline{D}_{s,(1,2)})} \geq \left(\sqrt{\frac{40}{9}} \approx 2.108\right)\right] \leq \epsilon \tag{31}$$

for a test with only one tail, since in this case we are only interested with one alternative, that $|\Delta \overline{D}_{s,(1,2)}| \neq 0$. Again, when the probability distribution function is Gaussian, the value of $\lambda$ for the one tailed case, would be $\approx 1.65$ instead of the $\approx 2.108$ demanded by Eq (26).

## Using the inequality to calculate significance when $\lambda \leqslant \sqrt{\frac{8}{3}}$

Theorem 1 does not hold, but you can still get an answer using Eq (26) as shown in the following Corollary 1 of the V-P inequality. We know propose and prove a corollary of Theorem 1:

**Corollary 1**. : *If $\lambda \leqslant \sqrt{\frac{8}{3}}$ then P is not known but is bound as*

$$\frac{1}{6} \leqslant P \leqslant 1 \tag{32}$$

*Proof.* As indicated by Eq (26), $\epsilon = \frac{4}{9\lambda^2}$ then if

$$\lambda \leqslant \sqrt{\frac{8}{3}} \Rightarrow \epsilon \geqslant \frac{4 \cdot 3}{9 \cdot 8} = \frac{1}{6} = 0.1\overline{6} = \epsilon_{th} \tag{33}$$

then $\epsilon_{th}$ is the value of $\epsilon$ at the threshold required for Theorem 1 to hold. When $\lambda \leqslant \sqrt{\frac{8}{3}}$ the predicted value of $\epsilon \geqslant \epsilon_{th}$, and since there is no limit to this inequality, it may happen that the estimated $\epsilon > 1$, but there is no probability $P > 1$, by definition. Thus when $\lambda \leqslant \sqrt{\frac{8}{3}}$ it follows that $\epsilon$ does not express a probability. But since if $\lambda \leqslant \sqrt{\frac{8}{3}}$ Eq (33) gives the highest $P$ value which may be estimated with the V-P inequality is $P = \frac{1}{6}$, $P$ is undetermined but bound in the interval:

$$\therefore \left(\lambda \leqslant \sqrt{\frac{3}{8}}\right) \Rightarrow \left(\frac{1}{6} \leqslant P \leqslant 1\right). \tag{34}$$

## Supporting information

**S1 Data.**
(XLSX)

## Acknowledgments

The authors thank Prof. Rafael Apitz (✉ rapitz@gmail.com) for its critical reading of the manuscript an his comments which helped to increase its clarity. This manuscript was written in LATEX using (**TEXstudio 4.1.1** (git 4.1.1-15-g0ff05de1) for Ubuntu GNU Linux, http://www. texstudio.org), an open source free LATEX editor with TEX Live 21 (https://www.tug.org/ texlive/). Most graphics were built using LibreOffice (v 7.2.3.1) Calc and combined into figures

using the GNU Image Manipulation Program v. 2.10.28 (GIMP). All these programs, free and open sourced, are available for Apple OS-X, *MicroSoft* Windows and GNU Linux.

## Author Contributions

**Data curation:** Antonio Javier Rodríguez-Hernández.

**Formal analysis:** Carlos Sevcik.

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
