## [Decision Letter · Decision Letter 0]

16 Oct 2021

PONE-D-21-14891Hidden chaos factors inducing random walks whichreduce hospital operative efficiencyPLOS ONE

Dear Dr. Sevcik,

Thank you for submitting your manuscript to PLOS ONE. After careful consideration, we feel that it has merit but does not fully meet PLOS ONE’s publication criteria as it currently stands. Therefore, we invite you to submit a revised version of the manuscript that addresses the points raised during the review process.

We look forward to receiving your revised manuscript.

Kind regards,

Sandra C. Buttigieg, MD PhD FFPH

Academic Editor

PLOS ONE

Journal Requirements:

2. In the Methods section of your manuscript, please clearly indicate that a waiver for ethics approval was obtained from the hospital IRB institute. Please also ensure that you have  include the full name of the ethics committee/institutional review board(s).

“All expenses were covered by Hospital La Fuenfría, Servicio Madrileño de salud (SERMAS). An non arbitered preliminary version of this article may be found in arXiv as http://export. arxiv.org/pdf/2011.09514. The authors thank Prof. Rafael Apitz for its critical reading of the manuscript an his comments which helped to increase its clarity”

“All funds were provided by Hospital La Fuenfría, Servicio Madrileño de Salud (SERMAS), Comunidad Autónoma de Madrid,  Madrid, Spain.”

Reviewers' comments:

Reviewer's Responses to Questions

**Comments to the Author**

1. Is the manuscript technically sound, and do the data support the conclusions?

Reviewer #1: Yes

Reviewer #2: Yes

2. Has the statistical analysis been performed appropriately and rigorously? 

Reviewer #1: Yes

Reviewer #2: Yes

3. Have the authors made all data underlying the findings in their manuscript fully available?

Reviewer #1: Yes

Reviewer #2: Yes

4. Is the manuscript presented in an intelligible fashion and written in standard English?

Reviewer #1: Yes

Reviewer #2: No

5. Review Comments to the Author

Reviewer #1: The paper is very interesting and well structured. The analysis conducted is appropriate and well communicated. The implications of the findings are also clearly outlined. However, the paper needs proof reading due to typos or missing punctuation; e.g. page 8 line 18 - These shows; page 8 line 20 and page 7 line 7 missing commas; page 9 line 40 - ''such as described'' should read ''such as those described..''

Reviewer #2: The aim of the authors is to show that factors not usually considered in hospital performance analysis may have a huge impact on hospital efficacy - by delving into the case of La Fuenfrĺa Hospital. The methodology is explained in a way that enables replication and the discussion and conclusions triangulate with the rest of the paper. However, the paper has many typos and grammatical mistakes, which makes it a difficult read and sometimes understand. Even the abstract seems like a literal translation from the original language and does not flow well. I suggest an complete proof read and edit of the article by a native English speaking person.

6. PLOS authors have the option to publish the peer review history of their article (what does this mean?). If published, this will include your full peer review and any attached files.

Reviewer #1: No

Reviewer #2: **Yes: **Prof. Simon Grima

---

## [Author Response · Author response to Decision Letter 0]

15 Dec 2021

Journal Requirements.

1. Please ensure that your manuscript meets PLoS ONE's style r ..... .

Reply: We did our best to comply. We used a published PLoS article to reproduce its form

information fields an and style. Not all the features were available in tha LaTeX remplate, but

we hope that any shortcomings will be easily solved for your typesetters. Also, most journal

use and require authos supplied Key Words, we added a small section with keywords to the

manuscript, if this is unsuitable it will be trivial for your typesetters to comment or erase the

Key Words section which is all in one line.

2. Has the statistical analysis been performed appropriately and rigorously? Reviewer #1:

Yes

Reviewer #2: Yes

Reply: No reply seems necessary.

3. Please update your submission to use the PLoS LaTeX template. ... ,

Reply: Done as suggested. This implied a long time modifying each of the entries of our

database to conform to PloS ONE uses on doi numbers and url links. It took a long time since

the BibTeX dbase contains 3315 entries and it most conform to the ABS book format we use for

a book citif 847 references, thus far.It was a lot of work. Perhaps the most important of our

references (Sevcik, 1998) appeared in an on line journal , Complexity International, which is

no longer available. Ten years ago we deposited a copy of Sevcik 1998 in arXiv, in a manner

that complies with the extinct Complexity International Journal copyrights. The link to the

arXiv file MUST be on the citation of this work, thus we had to tweak the citation to make the

arXiv ling visible even using PLoS One bibliography style.

4.

Thank you for stating the following in the Acknowledgments Section of your manuscript:

Reply: The information was set in the LaTeX template. It was divided into sections as done, for

example in https://doi.org/10.1371/journal.pone.0258052 which we use as styling example.

5. Please review your reference list to ensure that it is complete and correct. ...

Reply: Our references are stored in an extense BibTeX formatted data base used in all our

publications. However, we have to modify the fields thr references in our BiBTrX dbase to

accommodate to PloS One bibliography style without destroying the references format for other

uses.Reviewer's Responses to Questions

Comments to the Author

1. Is the manuscript technically sound, and do the data support the conclusions?

The manuscript must describe a technically sound piece of scientific research with data that supports

the conclusions. Experiments must have been conducted rigorously, with appropriate controls,

replication, and sample sizes. The conclusions must be drawn appropriately based on the data

presented.

Reviewer #1: Yes

Reviewer #2: Yes

Reply: No reply seems necessary.

2. Has the statistical analysis been performed appropriately and rigorously? Reviewer #1:

Yes

Reviewer #2: Yes

Reply: No reply seems necessary.

3. Have the authors made all data underlying the findings in their manuscript fully available?

Reply: We are unsure about the meaning of the YES choice by the reviewerss. It could mean in

the context of the question that we did it right in the manuscript, or that we should include

more. In any case we are including two MS Exel-formatted files with the original La Fuenfría

Hospital data. The manuscript, we feel, it is clear on how was the data used and corrected for

a short breech of missing data. In any event, we are open to any questions from the readers.

The original HLF spreadsheets provided have Spanish labels and headings, but each of these

may be easily translated to any language using Google Translate ® for example.

Reviewer #1: Yes

Reviewer #2: Yes4. Is the manuscript presented in an intelligible fashion and written in standard English?

Reply: As pointed out by Reviewer 2, there were some typos in the paper. We revised

once more, but since our capacity to detect mistakes in our papers decreases as we re-

read them, we hired a professional proofreader. We hope that now the issues are

fixed.

Reviewer #1: Yes

Reviewer #2: No

5. Review Comments to the Author.

Please use the space provided to explain your answers to the questions above.

Reviewer #1:

The paper is very interesting and well structured. The analysis conducted is appropriate and

well communicated. The implications of the findings are also clearly outlined. However,

the paper needs proof reading due to typos or missing punctuation; e.g. page 8 line 18 -

These shows; page 8 line 20 and page 7 line 7 missing commas; page 9 line 40 - ''such as

described'' should read ''such as those described..''

Reply: Thanks for your kind opinion. The revised version will be proof read by a

certified professional English proofreader. Prior to this, we re-read the manuscript

and corrected the typos we were able to see.

Reviewer #2:

The aim of the authors is to show that factors not usually considered in hospital

performance analysis may have a huge impact on hospital efficacy - by delving into the

case of La Fuenfría Hospital. The methodology is explained in a way that enables

replication and the discussion and conclusions triangulate with the rest of the paper.

However, the paper has many typos and grammatical mistakes, which makes it a difficult

read and sometimes understand. Even the abstract seems like a literal translation from the

original language and does not flow well. I suggest an complete proof read and edit of the

article by a native English speaking person.

6. PLoS authors have the option to publish the peer review history of their article (what

does this mean?). If published, this will include your full peer review and any attached

files.

If you choose “no”, your identity will remain anonymous but your review may still be

made public.Do you want your identity to be public for this peer review? For information

about this choice, including consent withdrawal, please see our Privacy Policy .

Reply: We appreciate Prof. Grima openness.

Reviewer #1: No

Reviewer #2: Yes: Prof. Simon Grima

---

## [Decision Letter · Decision Letter 1]

6 Jan 2022

Hidden chaos factors inducing random walks which

reduce hospital operative efficiency

PONE-D-21-14891R1

Dear Dr. Sevcik,

We’re pleased to inform you that your manuscript has been judged scientifically suitable for publication and will be formally accepted for publication once it meets all outstanding technical requirements.

Kind regards,

Sandra C. Buttigieg, MD PhD FFPH

Academic Editor

PLOS ONE

Additional Editor Comments (optional):

Reviewers' comments:

Reviewer's Responses to Questions

**Comments to the Author**

1. If the authors have adequately addressed your comments raised in a previous round of review and you feel that this manuscript is now acceptable for publication, you may indicate that here to bypass the “Comments to the Author” section, enter your conflict of interest statement in the “Confidential to Editor” section, and submit your "Accept" recommendation.

Reviewer #1: All comments have been addressed

Reviewer #2: All comments have been addressed

2. Is the manuscript technically sound, and do the data support the conclusions?

Reviewer #1: Yes

Reviewer #2: Yes

3. Has the statistical analysis been performed appropriately and rigorously? 

Reviewer #1: Yes

Reviewer #2: Yes

4. Have the authors made all data underlying the findings in their manuscript fully available?

Reviewer #1: Yes

Reviewer #2: Yes

5. Is the manuscript presented in an intelligible fashion and written in standard English?

Reviewer #1: Yes

Reviewer #2: Yes

6. Review Comments to the Author

Reviewer #1: I believe that the current version is suitable for publication. All comments raised by the reviewers have been addressed.

Reviewer #2: The author has successfully addressed the issues identified in a the first review wherein I had suggested minor review.

7. PLOS authors have the option to publish the peer review history of their article (what does this mean?). If published, this will include your full peer review and any attached files.

Reviewer #1: No

Reviewer #2: **Yes: **Simon Grima

---

## [Editor Report · Acceptance letter]

17 Jan 2022

PONE-D-21-14891R1 

Hidden chaos factors inducing random walks which reduce hospital operative efficiency. 

Dear Dr. Sevcik:

I'm pleased to inform you that your manuscript has been deemed suitable for publication in PLOS ONE. Congratulations! Your manuscript is now with our production department. 

Kind regards, 

on behalf of

Professor Sandra C. Buttigieg 

Academic Editor

PLOS ONE